# Magmatic underplating associated with Proterozoic basin formation: insights from gravity study over the southern margin of Bundelkhand craton, India

Ananya P. Mukherjee [1], Animesh Mandal [1]

[1]Department of Earth Sciences, Indian Institute of Technology, Kanpur, UP-208016, India

*Correspondence to:* Animesh Mandal (animeshm@iitk.ac.in)

**Abstract.** Extension tectonics responsible for intracratonic rift basin formation are often the consequences of active or passive tectonic regimes. The present work puts forth a plume-related rifting mechanism for the creation and evolution of two Proterozoic sedimentary basins outlining the Bundelkhand craton, namely the Bijawar and Vindhyan basins. Using global gravity data, a regional scale study is performed over the region encompassing the southern boundary of the Bundelkhand craton consisting of Bijawar basin, Vindhyan basin and Deccan basalt outcrops. The gravity highs in the central part of the complete Bouguer anomaly as well as the upward continued regional anomaly, derived from global gravity grid data, suggests that the Vindhyan sedimentary basin overlies a deeper high-density crustal source. The deepest interface as obtained from the radially averaged power spectrum analysis is observed to occur at a depth of ~30.3 km, indicating that the sources responsible for the observed gravity signatures occur at larger depths. 3D inversion of complete Bouguer anomaly data based on Parker-Oldenburg's algorithm revealed the Moho depth of ~32 km below the Vindhyan basin, i.e., south of the craton. 2D crustal models along two selected profiles showcase a thick underplated layer with maximum thickness of ~12 km beneath the southern part of the Bundelkhand craton. The inferred large E–W trending underplating and deciphered shallower Moho beneath the regions south of the exposed Bundelkhand craton points to crustal thinning compensated by magmatic emplacement due to a Paleoproterozoic plume activity below the craton margin.

## 1 Introduction

Plate tectonics involving rifting and convergence largely contribute to shaping the continental lithosphere. One of the driving forces behind these processes and mechanisms is associated with mantle plumes' interaction with the lithosphere. Such interactions modify the underlying crustal structure resulting in crustal thinning and magmatic emplacements as intrusive bodies within upper crustal layers and/or at the crust-upper mantle boundary. The magmatic bodies occurring at the base of the crust, known as underplating, play a significant role in crustal growth and evolution, thereby providing insights into the orogenies forming the current tectonic setup (Thybo and Nielsen, 2009; Thybo and Artemieva, 2013; Chouhan et al., 2020). Various tectonic settings, like rift basins, collisional zones, volcanic provinces, and cratons affected by plumes, are attributed to the presence of underplated layers. The connection between plumes and plate tectonics in the growth and break-up of supercontinents has been explored by numerous studies, like, Thybo and Artenieva (2013), Gerya (2014), Gerya et al. (2015), Puchkov (2016), Chen et al. (2020), Niu (2020), Melankholina (2021), and Ray et al. (2023). Extension tectonics can be associated with rifting either at far-off continental margins or initiated by uplift due to an upwelling mantle plume. The formation of intracratonic rift basins is generally credited to such extension tectonics, often accompanied by magmatic activities and formation of depressions, hosting sedimentary sequences deposited in different environments interlayered with volcanic formations, that cause underplating at the crust-mantle boundary (Thybo and Nielsen, 2009; Thybo and Artemieva,

2013; Chouhan et al., 2020). The process of underplating leads to the formation of materials with high density and high magnetic susceptibility properties at the deep crustal levels. Such a process also aids in the formation of the low-density continental crust by magma fractionation during the Earth's early evolution history (Kumar et al., 2012; Thybo and Artemieva, 2013).

The Proterozoic sedimentary basins of India preserve the imprints of tectonics and records of crustal reworking experienced by underlying crust and surrounding cratonic landmasses, providing insights into the processes involved in restructuring of the crust below the associated cratons and adjoining areas. The Proterozoic age Bijawar basin and Vindhyan basin sequences lie along the southeastern and southern margins of the exposed Bundelkhand craton (Fig. 1). Their formation initiated during extensional tectonics and the subsidence of Vindhyan basin continued through the later collisional processes between the Bundelkhand craton landmass and the southern Indian landmass as the age of their formation is constrained between ~2.0−1.6 Ga (Chaturvedi et al., 2012; Basu and Bickford, 2015; Chakraborty et al., 2015; Mishra, 2015; Rawat et al., 2018; Chakraborty et al., 2020; Colleps et al., 2021). The opening of the Bijawar basin, though constrained at Paleo-Mesoproterozoic ages, is still uncertain in terms of the geodynamic processes initiating the rifting of the stable cratonic landmass (Colleps et al., 2021). Several authors have assessed and proposed different geodynamic models depicting the mechanisms responsible for the development of the Bijawar basin (Malviya et al., 2006; Chaturvedi et al., 2012; Pandey et al., 2012; Chakraborty et al., 2015; Meert and Pandit, 2015; Mishra, 2015; Chakraborty et al., 2020; Kumar et al., 2020; and Colleps et al., 2021) and subsequently the Mesoproterozoic aged Vindhyan basin (Bose et al., 2001; Ray et al., 2002; Ray et al., 2003; Sarangi et al., 2004; Mishra and Kumar, 2014; Mishra, 2015; Colleps et al., 2021). According to these researchers, the formation of these basins is associated with the break-up and assembly of supercontinents like the Columbia and Rodinia, which hosted the Bundelkhand cratonic landmass through the geological past. Yedekar et al. (1990) proposed southward subduction of the Bundelkhand craton under the Bastar craton. Later, Roy and Prasad (2003) interpreted a northward subduction of the Bastar craton under the Bundelkhand landmass. Kumar et al. (2012) developed a shear velocity structure beneath the Archean Bundelkhand craton and the Proterozoic Vindhyan basin to validate the view that the Archean crust is less mafic than the Proterozoic crust, owing to the presence of a mafic layer underlying the latter's crustal layers. Gokarn et al. (2013) used the magnetotelluric method and resistivity information to observe that the Bundelkhand craton does not extend beneath the Vindhyan basin sequences towards the south. Previous gravity studies conducted in and around the Bundelkhand craton area (Tiwari et al., 2013; Mishra and Kumar 2014; Mishra, 2015; Kumar et al., 2020), have observed gravity high anomaly over the regions south of the Bundelkhand craton and the Vindhyan basin. This long wavelength high gravity anomaly encompasses the seismic stations studied by Kumar et al. (2012), namely Allahabad, Rewa and Sagar, indicating the possible influence of the deep crustal mafic layer on the gravity signatures. Prasad et al. (2022) estimated the Curie depth ranges, utilizing aeromagnetic and satellite magnetic data of the central Indian shield, of Vindhyan basin and Bundelkhand granitic massif as 26−40 km and 29−42 km, respectively. A recent study by Pavankumar et al. (2023) delineated the electrical Moho below the Bundelkhand craton by conducting a magnetotelluric survey in the northeast region of the craton, which highlighted the moderately conducting upper mantle beneath the craton.

Mishra (2015) had suggested a plume/superplume setting responsible for the formation of the Bijawar and Mahakoshal basins as interior rift and marginal basins, respectively, with respect to the Bundelkhand craton. The emplacement of Paleoproterozoic (~1.98−1.97 Ga) mafic sills within the intracratonic Bijawar basin also suggests the role of plumes in their origin (Singh et al., 2021). This plume/superplume concept could possibly be linked with the proposed mafic layer within the crust below Vindhyan basin evidenced in the studies performed by Kumar et al. (2012). Other researchers have also mentioned the existence of a mafic underplated layer below the region covered by the extensive Vindhyan basin (e.g., Malviya et al., 2006; Chaturvedi et al., 2012; Pandey et al. 2012; Chakraborty et al. 2015; Meert and Pandit 2015; Kumar et al. 2020; Colleps et al., 2021). A subsurface model depicting the spatial and depth extent of the underplated layer based on

geophysical observation and its correlation to the development of the Proterozoic basins along the southern margin of the Bundelkhand craton and adjoining areas is lacking. Whether the extensional process that initiated the formation of the Bijawar basin and the later evolution of the Vindhyan basin is due to a plume/superplume located below the Bundelkhand cratonic crust or was an effect of passive stretching at far-off plate margins is also still unclear.

The possible processes responsible for the subsidence aiding the Vindhyan supergroup formation range from either an extensional setup for the deposition of the Lower Vindhyan series (1.7−1.6 Ga), followed by the Upper Vindhyan series (1.1−0.7 Ga), in the form of an intracratonic basin, or as a large foreland basin accompanying the convergence of the Bundelkhand landmass along the Satpura Mobile Belt with the Bhandara-Bastar landmass (Bose et al., 2001; Ray et al., 2002; Ray et al., 2003; Roy and Prasad, 2003; Sarangi et al., 2004; Mishra and Kumar 2014; Mishra, 2015; Colleps et al., 2021; Mohanty, 2023). Colleps et al. (2021) provided the age constraints for the Lower Vindhyan Semri Group by studying the detrital zircon and suggested that the foreland basin may not be an appropriate model for the Vindhyan basin evolution during the deposition period of the Lower Vindhyan series. Creation of a foreland basin due to subsidence requires a prolonged orogeny which is not evidenced during the deposition of the Proterozoic basins of India (Basu and Bickford, 2015). This could indicate the role of upwelling mantle material which further facilitates the rifting of continental blocks, giving rise to the basins, crustal thinning that is compensated by underplated mafic material within the crust below such basins.

Satellite derived global free-air gravity and topography data are used to decipher the crustal configuration beneath regions lying around the southern margin of Bundelkhand craton (Fig. 1) and delineate the extent of the plausible underplated layer below the areas consisting of the exposed southern boundary of the craton flanked by the Bijawar group of rocks, Vindhyan basin outcrops and the Deccan traps. Parker-Oldenburg's 3D gravity inversion and 2D forward modelling approaches are utilized to compute the Moho structure and crustal configuration to illustrate the cause of the high regional gravity anomalies as observed in the gravity anomaly maps of the study area. The inversion algorithm deciphered a shallow Moho structure, suggesting crustal thinning below the area outlining the exposed Vindhyan rocks and their contact with the Bundelkhand craton. The forward models obtained along two profiles, (Fig.1) using the complete Bouguer anomaly data, depict the presence of a high-density crustal source at the base of the crust, spanning the area beneath the Vindhyan basin sequences. The depth to the top of this high-density layer correlates with the shallow Moho topography as observed in the inverted Moho depth map. The study illustrates the crustal structure below the areas adjoining the exposed southern margin of the Bundelkhand craton, showing the presence of the underplated mafic layer and provides evidence to further examine the concept of plume/superplume responsible for the formation of the Proterozoic basins bordering the craton. Thus, the study offers insights into the formation mechanism of the intracratonic rift basins like Bijawar and Vindhyan basins along the southern margin of the Bundelkhand craton.

## 2 Geological background

The Bundelkhand and Aravalli cratons in the northern part of peninsular India are separated from southern peninsular India (consisting of the Bastar and Dharwar cratons) by the Central Indian Tectonic Zone (CITZ) (Fig. 1a), (Roy and Prasad, 2003; Meert et al., 2010; Mishra, 2011; Harinarayana and Veeraswamy, 2014; Podugu et al, 2017; Chattopadhyay et al., 2020; Pati, 2020; Dessai, 2021). The craton is bordered by the Son-Narmada fault in the south and overlain by the Indo-Gangetic alluvium plain to the north of the Bundelkhand Tectonic Zone (BTZ) (Pal and Kumar, 2019; Mandal et al., 2020; Dessai, 2021). The exposed Bundelkhand craton is flanked on its eastern, western, and southern sides by the Vindhyan Supergroup (Fig. 1b). The major lithology of the Bundelkhand craton comprises Archean tonalite-trondhjemite-granodiorite (TTG) gneisses, volcano-sedimentary rocks, meta-supracrustals (amphibolites, komatiitic basalts), the Madawara ultramafic complex (MUC), Bundelkhand granitoid, along with quartz reefs and mafic dyke swarms across the craton (Slabunov et al.,

2017; Pati et al., 2019; Ramiz et al., 2019). The study area, located between 78° E − 80° E and 23°30' N − 25° N for the present work, comprises the southern part of the Bundelkhand craton along with the Vindhyan basin rocks outlining the southern craton boundary (Fig. 1b). Exposures of Deccan traps occur around the southwestern margin of the Bundelkhand craton, while the southeastern edge shows exposures of the Bijawar group of rocks (Fig. 1b) (Podugu et al., 2017; Pal and Kumar, 2019; Pati, 2020). The Bijawar group largely consists of volcanogenic metasediments with major basic/ultrabasic

intrusions supposedly formed in a rift environment over the rifted platform of the Bundelkhand craton (Crawford, 1970; Sarkar et al. 1984; Mondal et al. 1998; Mishra, 2014; Mishra, 2015). Rocks belonging to the Bijawar basin now form the base of the Vindhyan basin (Basu and Bickford, 2015; Mishra, 2015). Only parts of its rock sequences are exposed along the southern margin of the Bundelkhand craton (Fig. 1b) and with the Bundelkhand craton form the basement of the Vindhyan sediments of Paleo-Neoproterozoic time (Basu and Bickford, 2015; Mishra, 2015; Colleps et al., 2021). The

Vindhyan supergroup comprises the Semri (Lower Vindhyan), Kaimur, Rewa and Bhander (together form the Upper Vindhyan) series, consisting primarily of sandstones, limestones, and shales (Ray et al., 2002; Ray et al., 2003; Sarangi et al., 2004; Mishra, 2015).

The study area also comprises NW-SE trending mafic dyke swarms (1.1−1.97 Ga) (Pati, 2020) and NE-SW trending quartz reefs (1.9−2.0 Ga) (Pati et al., 2007; Pradhan et al., 2012; Bhattacharya and Singh, 2013; Pati, 2020). The trends of the quartz

reefs and the dyke swarms are correlated with the direction of the rifting process corresponding to the opening of the Bijawar basin in the Paleoproterozoic times (Mishra, 2015). Slabunov and Singh (2022) suggested that the swarm of giant quartz veins associated with the Bundelkhand craton points to the deformation undergone by the craton due to the collision processes and plume activity related to the Columbia supercontinent. The geological evolution of the landmass comprising the present study area ranges from around 3.5 Ga up to approximately 1.0 Ga (Basu and Bickford, 2015; Chakraborty et al.,

2015; Ramiz et al., 2019; Pati, 2020; Colleps et al., 2021). Throughout evolution, this region is said to have undergone several phases of tectonic activity that resulted in the formation of the Bijawar and Vindhyan basins. Different modes of formation are put forth by various authors, like, polyphase tectonic evolution of Bijawar basin (Chaturvedi et al., 2012), rift-related tectonics (Chakraborty et al., 2015), plume-related genesis of the basin (Singh et al., 2021), formation of the Satpura Orogeny (~2.2 Ga) leading to the formation of the Paleoproterozoic basins like Bijawar basin (Mohanty, 2023), intracratonic

rift basin formation (Basu and Bickford, 2015; Mishra, 2015), foreland basin formation due to subsidence of Vindhyan basin (Chakrabarti et al., 2007).

## 3 Data and methodology

### 3.1 Global gravity grid data

The regional scale study encompassing the southern part of the Bundelkhand craton and areas lying along the craton's

southern boundary is carried out by utilizing improved high-resolution free-air gravity anomaly grid data and topography data from the website of the Scripps Institution of Oceanography (https://topex.ucsd.edu/cgi-bin/get_data.cgi). This global gravity model of 1-minute grids has approximately 2 mGal accuracy and is based on data from the Geosat and ERS-1 satellites, along with new altimeter data from Jason-1 and CryoSat-2 satellites (Smith and Sandwell, 1997; Sandwell et al., 2014; Kende et al., 2017). The topography map for the study area (Fig. 2), showing a variation from 175 m to 617 m, derived

from the global topography grid (Smith and Sandwell, 1997). The acquired free-air anomaly and the topographic data are used to calculate Bouguer anomaly data by applying the Bouguer plate correction considering 2670 kg/m$^3$ as the average crustal density. The 'Terrain correction' module in the Gravity menu, available on the Oasis Montaj Geosoft software was used to obtain the terrain correction, which was then applied to the Bouguer corrected gravity anomaly. The maximum value of terrain correction obtained is ~ 0.93 mGal. The obtained terrain corrected Bouguer anomaly is gridded using the minimum

curvature interpolation technique and plotted as the complete Bouguer anomaly map (Fig. 3). The maximum and minimum gravity values as calculated from the global gravity data, are ~ −32.1 mGal and ~ −67.3 mGal.

## 3.2 Regional- residual separation

The complete Bouguer anomaly (Fig. 3) is a combination of the signals due to both deeper sources and shallow-level features, known as the regional anomaly and residual anomaly, respectively. For identifying the effects of both these sources of gravity
signatures independently, the two anomalies need to be separated. The upward continuation method (Pacino and Introcaso 1987; Blakely 1995) is utilized here to obtain the larger wavelength anomalies corresponding to the deeper source of the gravity anomalies. The choice of the upward continuing heights was based on a trial-and-error approach as suggested by Gupta and Ramani (1980). This regional anomaly when subtracted from the complete Bouguer anomaly yields the residual anomaly. In this study, we have applied this method to obtain the regional gravity anomalies using upward continuing heights
of 60 km, 30 km, and 10 km (Fig. 4a, 4c and 4e). The upward continued regional gravity anomalies show highs occurring in the SW corner (Fig. 4a, 4c, and 4e, respectively). The obtained regional gravity anomaly maps show the variations and trends observed due to deep-seated features and source bodies. These regional maps are further used to remove the longer wavelength signatures from the complete Bouguer anomaly map to get the respective residual gravity anomalies (Fig. 4b, 4d, and 4f). The trends of the high and low anomalies seen in residual maps obtained from the 60 km, 30 km as well as the
10 km upward continued regional anomalies (Fig. 4b, 4d, and 4f) show correlations with the observed geological units of the study area. The gravity highs in the residual gravity anomaly maps correspond to the southwestern Deccan outcrops and Bijawar basement below the Vindhyan sequences along the southern craton margin, while the gravity lows relate to the Bundelkhand granitic complex, covering much of the study area in the north as well as the thick Vindhyan sedimentary formations towards the south of the study area (comparing Fig. 1b and Fig. 4b, 4d, and 4f).

## 3.3 Depth estimation using the radially averaged power spectrum (RAPS) method

Depths to the tops of the subsurface geologic features, intrusions, and the basement complex can be deciphered by utilizing the radially averaged power spectrum (RAPS) technique based on spectral analysis of calculated gravity anomaly in the Fourier domain (Spector and Grant, 1970; Saada, 2016; Mandal et al., 2020). First, the complete Bouguer anomaly is transformed using the Fast Fourier Transform, and then the 'Spectrum Calculation and Display' feature under the MAGMAP
menu of Geosoft, calculates the radially averaged power spectrum from the complete Bouguer anomaly data. The plot shows the natural logarithm of power of the respective anomalies against wavenumbers. The smaller wavenumber values correspond to the information from the deeper sources, while the larger wavenumbers depict the shallow surface sources. The average depths to the tops of various sources are estimated by finding the slopes of line segments drawn through few consecutive points on the plot, and then dividing the slope by $-4\pi$. The RAPS plot along with the depth estimates as obtained
from the complete Bouguer anomaly values is shown in Fig. 5.

## 3.4 Three-dimension gravity inversion for Moho topography

The Moho structure below the proposed study area is computed using a MATLAB-based program developed by Gomez-Oritz and Agarwal (2005) and further modified by Gao and Sun (2019) following the Parker-Oldenburg algorithm. This algorithm uses a Fast Fourier transform (FFT) based forward (Parker, 1972) and inverse (Oldenburg, 1974) gravity modeling
scheme in three dimensions. The iterative inversion method calculates the gravitational field due to the Moho interface modelled using an assumed mean depth, $z_0$ and density contrast, $\Delta\rho$. The values of $z_0$ and $\Delta\rho$ are chosen with respect to the prior geological and geophysical knowledge of the study area to reduce the ambiguities in the obtained models. Mapping the Moho interface using this approach has been carried out by several workers (e.g., Gomez-Oritz and Agarwal, 2005;

Meijde et al., 2013; Windhari and Handayani, 2015; Abdullahi et al., 2019; Bessoni et al., 2020; Chen and Tenzer, 2020; Ydri et al., 2020).

Parker (1972) first derived the expression of the vertical component of the gravity anomaly, $\Delta g(x)$ due to an undulating interface, in Fourier domain as,

$$F[\Delta g(x)] = -2\pi G \Delta \rho \, e^{-|\bar{k}|z_0} \sum_{n=1}^{\infty} \frac{|\bar{k}|^{(n-1)}}{n!} F[h^n(x)],$$
(1)

where, $F[\Delta g(x)]$ is the Fourier transform of the gravity anomaly, $G$ is the gravitational constant, $\Delta \rho$ is the density contrast across the interface, $|\bar{k}|$ is the wavenumber, $h(x)$ is the depth to the concerned interface (the depth increases downwards) and $z_0$ is the mean depth of the interface. Equation (1) was reorganized by Oldenburg (1974) to iteratively compute the depth to the interface, i.e., the undulating Moho discontinuity, from the gravity anomaly using the equation,

$$F[h(x)] = -\frac{F[\Delta g(x)]e^{|\bar{k}|z_0}}{2\pi G \Delta \rho} - \sum_{n=2}^{\infty} \frac{|\bar{k}|^{(n-1)}}{n!} F[h^n(x)].$$
(2)

Gao and Sun (2019) re-derived the Eq. (1) taking the vertical z-axis as positive downwards, unlike Gomez-Oritz and Agarwal (2005). A simplified expression for the modified Eq. (1) by Gao and Sun (2019) can be written as,

$$F(\Delta g) = 2\pi G \Delta \rho \, e^{-|\bar{k}|z_0} \sum_{n=1}^{\infty} \frac{|\bar{k}|^{(n-1)}}{n!} F[(-h)^n(x)].$$
(3)

With the revised Eq. (3), the Eq. (2) is further rewritten in the simplified form (Gao and Sun, 2019) as follows,

$$F[-h(x)] = \frac{F[\Delta g]e^{|\bar{k}|z_0}}{2\pi G \Delta \rho} - \sum_{n=2}^{\infty} \frac{|\bar{k}|^{(n-1)}}{n!} F[(-h)^n(x)].$$
(4)

Gao and Sun (2019) modified the original algorithm given by Gomez-Oritz and Agarwal (2005) using the formulae shown in Eq. (3) and Eq. (4) and the present study uses this modified algorithm to compute the inverted Moho topography and the gravity anomaly resulting from the calculated Moho interface.

Gomez-Oritz and Agarwal (2005) observed that the inversion process using Eq. (2) is highly unstable for data with high frequencies (larger wavenumbers). This requires a high-cut filter to facilitate the convergence of the inversion process. This also follows that the interface of interest lies at larger depths and hence the focus would be on smaller frequency data. Thus, a high-cut filter is introduced (Gomez-Oritz and Agarwal, 2005). $WH$ is the minimum wavenumber and $SH$ is the maximum wavenumber, which allows only lower frequency data (for smaller $k$ values) and eliminates the high frequency data of the complete Bouguer anomaly to be used for the algorithm, defined as,

$$HCF(k) = \frac{1}{2}\left[1 + \cos\left(\frac{2\pi(k-WH)}{2(SH-WH)}\right)\right].$$
(5)

For $WH<k<SH$

$HCF(k)=0$ for $k>SH$

$HCF(k)=1$ for $k<WH$

The complete Bouguer anomaly values of the global gravity grid data used for the inversions is provided in ASCII format as the input in a square grid of $222\times222\,\text{km}^2$, with a total of $120\times120$ columns and rows. To minimize any edge effects, the Turkeywin function is used (Gomez-Oritz and Agarwal, 2005; Windhari and Handayani, 2015). The mean depth, $z_0$ and density contrast, $\Delta\rho$ between two crustal layers are logically selected to calculate the topography of the Moho interface. The inversion algorithm was performed for obtaining the Moho topography using varying values of mean Moho depth ($z_0$), ranging from 30 km to 38 km, at 2 km intervals. The chosen range of mean Moho depth values are based on the deepest depth estimate from RAPS analysis (Fig. 5) and based on prior literature (e.g., Kumar et al., 2012), respectively. The density contrast is taken to be $520\,\text{kg/m}^3$, considering the mantle density to be $3300\,\text{kg/m}^3$, and the average crustal density to be

2780 kg/m³, (using the data from Table 1, including the density of the proposed crustal underplated layer as 3150 kg/m³). The inverted Moho topography results calculated by using the different mean Moho depth values, with a constant density contrast, indicate that the derived Moho depths are sensitive to a 2 km variation in the $z_0$ values. Further, a better correlation with the crustal layer thicknesses and depths suggested by Kumar et al. (2012) is observed for the calculation of Moho topography considering mean Moho depth as 36 km. The values of the cut-off parameters, WH and SH, are taken as 0.01 km$^{-1}$ and 0.012 km$^{-1}$, respectively. These values correspond to 100 km and 83.33 km wavelength data respectively, associated with long wavelength information for the Moho interface.

The gravity data is demeaned first and then an amplitude spectrum, along with a matrix of the corresponding frequencies is computed using fast Fourier transform. The iterative process then begins, the first term is calculated using Eq. (3) and the obtained topography in wave number domain is filtered using the HCF filter. Inverse fast Fourier transform is applied to compute the topography in space domain. The newly obtained topography is then used to compute the second term using the Eq. (4), which is again filtered, and a new topography of the interface is computed with inverse fast Fourier transform. This iterative procedure is continued until convergence is reached. The convergence criterion used for this study is 0.02 km (Gomez-Oritz and Agarwal, 2005), i.e., the iteration process stops once the RMS error between the new topography and the previously calculate one is lower than the convergence criterion. The outputs, obtained after the iteration procedure is over, are the inverted topography (Fig. 6a), the gravity due to the inverted topography (Fig. 6b), the difference between the input gravity and the output gravity, number of iterations taken and the final RMS values. The relief of the interface must be less than the assumed mean depth of the interface (Gomez-Oritz and Agarwal, 2005).

## 3.5 Two-dimension forward gravity modelling

The complete Bouguer anomaly derived using the global grid data is utilized to generate 2D crustal models across the profiles AA' and BB' as shown in Figure 1. The profiles are chosen to determine the crustal structure under the areas encompassing the contact between Bundelkhand craton and Vindhyan basin, along with parts of Deccan trap exposures. The GM-SYS profile module of the Oasis Montaj software is used for performing the 2D forward modelling along the two profiles. The 2D forward modelling responses are based on the methods of Talwani et al. (1959), and Talwani and Heirtzler (1964), that make use of the algorithms given by Won and Bevis (19870) (Oasis Montaj GM-SYS user's guide). Information of geological units composing the study area, along with density values have been considered based on the works of Basu and Bickford (2015), Meert and Pandit (2015), Mishra (2015), Podugu et al. (2017), Pal and Kumar (2019), Pati and Singh (2020), Colleps et al. (2021). Density variations of the surface lithology and the crustal layers are also utilized from prior geophysical studies over the CITZ and Aravalli-Delhi Mobile belt (ADMB) (Rao et al., 2011; Mishra and Kumar, 2014; Mishra, 2015). The thickness values of the different layers are constrained using information from the studies conducted using wide-angle seismic method along the Hirapur-Mandla profile of Sain et al. (2000) and the shear velocity structure beneath the Sagar (SGR, Moho mapped at ~44 km) seismic station by Kumar et al. (2012). Thus, the 2D forward models are developed utilizing the above-mentioned literature, along with the exposed lithology information (Fig. 1b), as well as the depths and crustal layer information as obtained from RAPS analysis (Fig. 5), and inverted Moho topography (Fig. 6a). The average density values used for various litho-units of the study area to generate effective crustal models along the profiles are given in Table 1.

## 4 Results

### 4.1 Gravity anomaly

The complete Bouguer anomaly map (Fig. 3) of the global gravity grid data shows a centrally located and mostly E-W trending high gravity anomaly region (−47 to −29 mGal) coinciding with the southern boundary of Bundelkhand craton. The 60 km, 30 km, and 10 km upward continued regional anomalies, all show the similar E-W trending gravity high (Fig. 4a, 4c, and 4e, respectively). The southwestern corner of the complete Bouguer anomaly map also shows gravity high signatures correlating with the exposures of the Deccan traps as seen in the geological map (Fig. 1b). The corresponding residual gravity anomaly maps obtained for each of the upward continued regional gravity anomaly map also show correlations with the trends of the lithological units observed in the geological map (Fig. 1b). The 60 km upward continued regional gravity anomaly shows high gravity signatures in the southwestern corner, decreasing towards the northeastern corner, with high to moderate values in the central part of the study area (Fig. 4a). The regional and residual anomaly maps obtained by the 10 km upward continuation method (Fig. 4e and 4f, respectively) show the centrally located gravity high similar to that obtained from the 30 km upward continuation method (Fig. 4c and 4d, respectively). These suggest the presence of high-density sources at deeper as well as shallower depths. This central region of the study area is covered by rocks belonging to the Vindhyan Supergroup (Fig. 1b), with possibly a high-density basement along with the Bundelkhand granitic basement. The low anomaly seen in the bottom right-hand corner of the complete Bouguer anomaly (Fig. 3) is associated with the thick sedimentary formations of the Vindhyan basin. The high anomalies seen in the southwestern corner of the residual gravity anomaly maps correlate with the outcrops of the Deccan traps lining the Vindhyan sedimentary basin (comparing Fig. 1b with Fig. 4b, 4d, and 4f). The gravity highs, appearing in the complete Bouguer anomaly map (Fig. 3) as well as the regional-residual maps obtained by upward continuing the complete Bouguer anomaly data up to 60 km, 30 km, and 10 km (Fig. 4), over the regions covered by the thick sedimentary sequences of the Proterozoic Vindhyan basin provided the motivation to identify the depth and nature of the sources giving rise to such gravity signatures. The low anomaly values seen towards the top of the complete Bouguer anomaly map, and the regional gravity anomaly maps correlate with the Archean age Bundelkhand gneissic complex of the craton (comparing Fig. 1b with Figs. 3, 4a, 4c, and 4e, respectively).

### 4.2 Depth estimates by radially averaged power spectrum (RAPS) analysis

The RAPS analysis of the complete Bouguer anomaly from global gravity data indicates the depths to the top of three interfaces at ~30.3 km, ~11.9 km and ~2.7 km (Fig. 4), suggesting the existence of deeper sources. The regional-residual gravity anomaly maps based on the upward continuation heights of 60 km, 30 km, and 10 km (Fig. 4a, 4c, 4e, respectively) correlate with the depth estimates from radially averaged power spectrum analysis, i.e., ~30.3 km, ~11.9 km, and ~2.7 km, respectively. Thus, it can be interpreted that the observed E-W trending high to moderate gravity signature in the central section of the regional gravity anomaly map obtained from 60 km upward continuation (Fig. 4a) is probably due to sources located at more than ~30 km below the surface, corresponding to the deepest depth estimate from the RAPS plot (~30.3 km, Fig. 5). This indicates high density sources lying approximately at depths between the lower crust and mantle. The centrally located high gravity anomaly, as seen for both 30 km and 10 km upward continued regional gravity anomalies (Fig. 4c, 4e, respectively), exhibit high gravity signatures due to high-density material are observed at depths shallower than 30 km, closely correlating to the depth estimates of ~11.9 km, and ~2.7 km from the RAPS plot (Fig. 5). These suggest the continuation of high-density sources at deeper as well as shallower depths.

### 4.3 Inverted Moho topography

The contour maps of the Moho topography and corresponding gravity anomaly (Fig. 6a and 6b, respectively, with the study area marked as a red box) are plotted based on the results obtained from the MATLAB based algorithm described in section 3.4. The inversion process was completed in three iterations, giving an RMS error value of 0.0121 km, which is less than the assigned convergence criterion (0.02). The maximum Moho depth of 44 km is obtained over the low density Bundelkhand granitoid complex (Figs. 1b and 6a) in the northern part of the study area. The minimum Moho depth is estimated to be 32 km over the central part of the study area covered by Vindhyan basin sequences (Figs. 1 and 6a). The moderate to shallow Moho depth variation observed in the southwestern corner of study area marked on the Moho depth map (Fig. 6a) correlates with the occurrence of the Deccan traps (Fig. 1b). The calculated gravity anomaly values due to the inverted Moho topography range from a minimum of −30 mGal to a maximum of 20 mGal (Fig. 6b). The lowest values are observed where the obtained Moho depth is highest, i.e., below the northern study region, and the high gravity values are situated over the area with shallow Moho, i.e., beneath the central region of the study area (see Fig. 6).

### 4.4 Two-dimensional forward modelling

Forward modelling of the complete Bouguer anomaly (Fig. 3) from the global grid data is performed along two profiles (namely, AA' and BB', Fig. 1b) spanning across the contact between the Bundelkhand craton and the sedimentary Vindhyan basin along the craton's margin using Oasis Montaj Geosoft software as described in section 3.5. These models provide better insights into the extent of the high-density crustal source and the crustal structure beneath the study area, thereby also giving a way to verify the results of Moho topography obtained by the gravity inversion algorithm. The models are constrained by the exposed geological information, density information (see Table 1), crustal thickness information, and geodynamic setups as discussed in previous sections. The depth extents are further adjusted utilizing the RAPS depth estimates, the Moho depths from the inversion algorithm as well as the layer thickness information from the SGR seismic station of Kumar et al. (2012). The density and structure of the underplated are adjusted by a trial-and-error approach, with support from Kumar et al. (2012) as well as the Moho inversion results.

The complete Bouguer anomaly response along profile AA' (Fig. 7) shows a central high and a low on the northwestern side of the profile (influenced by the Bundelkhand granites, gneisses) and a moderate low to the southeast of the profile. The density model shows the thickness of the Vindhyan basin rocks, upper crust and underplating along the profile AA' (Fig. 7). The high-density (3150 kg/m$^3$) underplating gains the maximum thickness of ~12 km in the central portion of the profile, almost directly below the rift basin structure consisting of the Vindhyan supergroup rocks and the high-density Bijawar group rocks, extending towards the southeastern corner. The layer thins out below the exposed Bundelkhand craton in the north of the profile. The Moho depth under this profile varies from ~39 km to ~42 km, shallowing up slightly below the cratonic area (Figs. 1 and 7).

The complete Bouguer anomaly along the profile BB' (Fig. 8) shows a high to the south and a low to the north of the profile. The southwestern part of the profile here shows a layer of Deccan traps of maximum thickness ~1.4 km. The depth to the upper mantle varies between 37 km and 40 km. The thickest part of the high-density underplating shows a thickness of ~4 km in the central part of the profile. The Moho in the southwestern part of this profile is slightly upwarped (~37 km), possibly due to the consequences of the extensive Deccan volcanism (~65 Ma) (Fig. 1b).

The deepest depth, ~30.3 km, attained from the RAPS plot (Fig. 4) of the complete Bouguer anomaly shows close correlation with the depth to the top of the underplating layer occurring at the base of the crust shown in these models. Comparing the models along the profiles AA' and BB', it is seen that the extent of the Bijawar rocks as the basement of the Vindhyan basin sequences decreases from the area near the Bundelkhand craton boundary towards the south and southeast, as well as southwest, towards the Deccan basalt exposures. This indicates that the Bijawar basin possibly narrows down towards the

southern and western direction along the southern boundary of the Bundelkhand craton. The high-density underplating layer is thickest in the central region (~12 km), decreasing the most towards the north, below the craton region, in both the profiles. The depth to the Moho varies between ~37 km to ~42 km, shallowing up below the Bundelkhand craton region and areas covered by Deccan traps. The underplating layer shows a central high along both AA' and BB' profiles. The Moho uplift, which is generally expected below areas affected by rifting processes, is instead compensated by the emplacement of the underplating above the Moho, as depicted by the 2D models (Figs. 7, and 8).

## 5 Discussion

The complete Bouguer (Fig. 3) and regional gravity anomaly maps (Fig. 4a, 4c, and 4e) illustrate the high-gravity signatures over the southern boundary areas of the Bundelkhand craton and the adjoining Vindhyan basin. The high gravity signatures in the residual gravity anomaly maps (Fig. 4b, 4d and 4f) as observed along the southern Bundelkhand craton boundary, do not seem to extend further south of the Bundelkhand craton margin (comparing Fig. 1b with Fig. 4b, 4d, and 4f). This implies that the regional gravity high observed in the 60 km and 30 km upward continued regional anomaly maps (Fig. 4a and 4c) is probably due to a high-density source with a large regional extent below the thick sedimentary Vindhyan sequences. The higher gravity anomalies in the central and southwestern regions, as observed in the regional as well as residual anomalies obtained from upward continuation heights of 60 km, 30 km, and 10 km, are due to sources at deeper and shallower depths (Fig. 4). The observed gravity highs in the central region of the complete Bouguer anomaly (Fig. 3) and regional anomaly maps (Fig. 4a, 4c and 4e) along with the geological set up of this region provide significant evidence to support the role of upwelling magma and eventual magmatic emplacement in the form of an underplated layer.

### 5.1 Crustal configuration around the southern boundary of Bundelkhand craton

The gravity signature due to Moho topography (Fig. 6b) reveals some similarity in trend with the 60 km and 30 km upward continued regional gravity anomaly maps (Fig. 4a and 4c, respectively). Thus, it can be inferred that the crust below the region immediately south of Bundelkhand craton covered by Vindhyan sedimentary basin exhibits a gravity response corresponding to a shallower crust compared to the cratonic regions. The 2D gravity models (Figs. 7 and 8) along profiles AA' and BB', respectively, leads us to interpret that the crust observed below the regions around the exposed southern Bundelkhand craton boundary hosts a ~2 to ~12 km thick mafic underplated layer above the Moho, which in turn is reflected as a shallower Moho in the inverted Moho depth map (Fig. 6a).

The gravity high in the southwest corner of all the upward continued regional as well as residual anomaly maps (Fig. 4) indicates the effects of the Deccan volcanic basalts lying in the region (Fig.1b). Shallow Moho depth and high density underplated layer often influence the gravity signatures giving rise to high gravity anomaly values (Chouhan et al., 2020). This supports that the vast Deccan volcanic activity probably influenced the crustal configuration of the adjoining Proterozoic Vindhyan basin region studied here. The influence of the Deccan volcanism can also be observed by the moderately high gravity values seen in the southwest corners of the complete Bouguer anomaly (Fig. 3). The effect of the emplacement of the Deccan basalts is seen as shallowing up of Moho interface in the Moho topography map (Fig. 6a) and in the southwest of the BB′ profile (Fig. 8), below the Deccan trap exposures. This suggests that crustal uplift may also be due to the extensive volcanic activity giving rise to the Deccan basalts at about 65 Ma (White and McKenzie, 1989) in the southwestern part of the present study area. The models computed in the present study gives an approximation of the crustal structure beneath the areas encompassing the southern part of the Bundelkhand craton and the areas covered by the Vindhyan and Bijawar basins along the exposed southern boundary of the craton.

## 5.2 Mafic underplating below Vindhyan basin

The observations made on Moho depth and Curie depth estimates from Kumar et al. (2012) and Prasad et al. (2022), respectively, indicate that the crust below the Proterozoic Vindhyan basin is thick and hosts deep crustal high-density and high magnetic susceptibility material. Mishra (2015) suggested the role of a large plume or superplume responsible for rifting between the then adjacent cratons, that supposedly provided the Bijawar marginal basin for deposition of sediments and wide scale mafic/ultramafic sequences, around ~2.0 Ga. Extension tectonics are often accompanied by magmatic activities leading to formation of rift basins subsequently filled by various forms of volcanic material and cause underplating at the crust-mantle boundary (Thybo and Artemieva, 2013). The occurrence of the mafic magmatism corresponding to the Paleoproterozoic craton-margin rifting process associated with the formation of Bijawar basin has been highlighted by Malviya et al. (2006), Chaturvedi et al. (2012), Pandey et al. (2012), Chakraborty et al. (2015), Meert and Pandit (2015), Mishra (2015), Kumar et al. (2020), Colleps et al. (2021), and Singh et al. (2021). The central high anomaly observed in the complete Bouguer anomaly map (Fig. 3) also suggests the presence of a high-density body with a large extent in the areas covered by the Proterozoic sedimentary basins. With the upward continued regional-residual gravity anomaly maps and computed 2D forward models, it can be sufficiently said that the highs observed is due to the volcanogenic sequences of Kurrat volcanics (Dar and Khan, 2016; Rawat et al., 2018) within the Bijawar group of rocks along with the thick high-density underplating emplaced in the lower crust just above the Moho. The underplating appears to be the thickest under the central regions of the profiles, lying along the contact between the Bundelkhand craton and the sedimentary basin sequences, as seen from the forward models of the two profiles (Figs. 7 and 8). This thick mafic layer at the base of the crust is also depicted by the shallow Moho topography in the inverted Moho depth map (Fig. 6a) below the central region of the study area.

The Moho depth variations in the Moho topography map (Fig. 6a), computed using the inverse program shows a range of depths from 32 km to 38 km for the areas covered by the Vindhyan lithology. This shallower depth variation could be the result of the limitation of the inversion code in differentiating the density contrast between the mafic layer and the upper mantle, which exhibits an upwarped Moho (Fig. 6a). Thus, the Moho depth map (Fig. 6a) also justifies the presence of the underplating/upwarped Moho which is a signature of the rifting conditions prevalent during the formation of Bijawar and Vindhyan basin. Thus, this map again reinforces the centrally located gravity high, which is prominent in the complete Bouguer anomaly and regional gravity anomaly maps (Figs. 3 and 4a, 4c, and 4e, respectively), is caused by the thick mafic underplated layer or due to Moho upwarping. There is a slight difference between the Moho depths below the Bundelkhand craton obtained from the 3D gravity inversion method and forward modelling technique. The inversion algorithm was performed while considering the underplated layer above the Moho as well as eliminating it for the assumed density contrast between the crust and the mantle. It was observed that a density difference of 150 kg/m$^3$ between the underplated layer (taking density as 3150 kg/m$^3$) and mantle (taking density as 3300 kg/m$^3$) is not very well distinguished by the algorithm. The resulting trend in the Moho depth variations obtained from the inversion do not appear to change significantly, irrespective of whether the underplated layer is considered in the average crustal density calculations or not while assuming the density contrast for the inversion. The density contrast (520 kg/m$^3$, with the density of the underplated layer, Fig. 6a) used for the inversion may not apply objectively for the region below Bundelkhand craton since the underplated layer is absent below the craton, as observed in the northeastern part of the 2D forward model along the profile BB' (Fig. 8). The thin (~1–3 km) underplating layer above the Moho of the cratonic areas in the forward crustal model along the profile AA' (Fig. 7) is difficult to be distinguished by the inverse method as discussed above, showing slightly different Moho than the 2D forward crustal model along profile AA'. While the 2D forward models indicate that the observed complete Bouguer anomaly is consistent with the presence of the proposed underplated layer overlying the Moho, there lies some limitation to the uniqueness of the obtained results from the forward modelling scheme. However, the similarity observed between crustal

configuration presented by the results discussed here and previous works by authors like Kumar et al. (2012), Mishra (2015), imparts a certain validity to the suggested evolution mechanism in this study.

## 5.3 Plume driven formation of Proterozoic basins

The geometry displayed by the Vindhyan supergroup and Bijawar basement rocks resembles a rift basin with a depth to the basement ranging from ~ 6 km to ~8 km (Figs. 7 and 8). These models justify the presence of the proposed large mafic bodies by Kumar et al. (2012) and Mishra (2015) with the focus in the regions lying around the contact between the Bundelkhand craton and the Vindhyan as well as Bijawar basins. The presence of such high-density magmatic material at the base of the crust points to the possible intracratonic rifting mechanism aiding the formation of the Bijawar basin, and eventually the Vindhyan basin. The contrast between the expected deeper Moho depths and the observed shallow Moho depth seen under the Vindhyan basin is due to the influence of the magmatic underplating at the base of the crust here, which takes the form of an upwarped Moho interface, as obtained from the inverted Moho topography (Fig. 6a). The Moho, as can be seen from the forward crustal models, (Figs. 7 and 8) is deeper than that observed in Moho depth map (Fig. 6a). The underplating layer modelled here appears to be extending further south, indicating their continued presence under the regions showing the exposed Vindhyan basin rocks. The observed high gravity anomaly values in the gravity map obtained from the inverted Moho interface highlights the presence of such high-density underplated layer below the Vindhyan basin region in the central portion of the study area (Fig. 6b, within the red box marking the study area). This corroborates with the previous studies proposing large mafic layer forming in the lower crustal parts of sedimentary basin formed by rifting processes and the crust generally being affected by large scale extension associated magmatic activities (Behera et al., 2004; Kumar et al., 2012; Thybo and Artenieva, 2013; Basu and Bickford, 2015; Chouhan et al., 2020; Singh et al., 2021).

The inferred underplating beneath the study area and its large E–W extent as shown by the models give an impetus to the proposed presence of plume/superplume below this region during the Paleo-Proterozoic times (Mishra, 2015). This plume probably was responsible for the rifting of the Bijawar basin and the consequent deposition of the Lower Vindhyan sequences (Patranabis-Deb and Saha, 2020; Colleps et al., 2021). With the aid of the interpretations from the developed models, we propose an evolution mechanism (Fig. 9) for the rifting of the Bijawar basin providing support to the plume/superplume related tectonic model as suggested by Mishra (2015). The presence of the plume, at ~2.5 Ga, underneath the Bundelkhand craton induced the extension responsible for the rifting of the Bijawar basin (Fig. 9a). The formation of the Bijawar rift basin (~2.2 Ga) was initially accompanied by crustal thinning as it is normally observed during rifting. Mohanty (2023) put forward that the northern Indian block came in proximity of the southern Indian block around this time (Fig. 9b). As the extension of the Bundelkhand landmass continued, the sediment supply to the Bijawar basin was generated as the erosional material from the Bundelkhand craton and got deposited on the shallow rifted platform (Chaturvedi et al., 2012; Colleps et al., 2021). Plume-related rifting of the Bijawar basin can be evidenced by the magmatic and volcanic sequences of the Bijawar Supergroup (Fig. 9b), namely, Dargawan sill and the Kurrat volcanics (Patranabis-Deb and Saha, 2020; Singh et al., 2021). The Lower Vindhyan group of rocks began getting deposited on the rifted platform of the Bijawar basin (~1.9 Ga), and the region was subjected to further extension due to the continued presence of the plume below the region (Fig. 9c). This plume can be attributed to the breakup of the supercontinent 'Columbia' as the age constraints of the related tectonic events (Mishra, 2015; Chakraborty et al., 2020; Slabunov and Singh, 2022) are approximately close to that of the plume activity described here.

The deposition of the Lower Vindhyan series halted at around ~1.5 Ga, and the onset of the Upper Vindhyan basin opening began thereafter (~1.4 Ga). The deposition of the Upper Vindhyan groups marks the convergence of the North Indian and South Indian landmasses along the CITZ (~1.1–0.7 Ga) (Fig. 9d) (Mishra, 2015; Patranabis-Deb and Saha, 2020). Based on the forward models presented in this study, it can be inferred that the crustal thinning due to rifting got compensated by the

emplacement of the high-density material above the Moho in the subsurface part of the extended region due to the existing plume, like a failed rift basin structure as documented by Thybo and Nielsen (2009). Mishra (2015) proposed the plume/superplume tectonics based on just the presence of high-density Bijawar basement, the present study strengthens this hypothesis by delineating the underplated layer associated with such plume related rifting environments. The plume which was responsible for the extension previously could have also facilitated the down thrusting of the Bundelkhand basement

below the South Indian block (Fig. 9d), leading to the N-S collisional event between the Bundelkhand craton and the southern (Bastar, Bhandara and Dharwar) cratons along the CITZ. This convergence and the consequent closing of the Vindhyan basin holds significance in the assembly of the 'Rodinia' supercontinent, which supposedly existed at ~1 Ga (Roy and Prasad, 2003).

**6 Conclusions**

The complete Bouguer anomaly using the global grid gravity data highlights the large-scale E–W trending, centrally located high anomaly, encompassing the areas covered by southern Bundelkhand craton as well as the adjoining Deccan traps and Vindhyan basin, further south of the exposed southern boundary of the craton. The 60 km and 30 km upward continued regional gravity anomaly maps, along with the depth estimates from the radially averaged power spectrum plot suggest a deep crustal, high-density source below this region giving the E-W trending high gravity anomaly. The inverted topography

of the Moho interface computed using the complete Bouguer anomaly and the corresponding gravity anomaly obtained using the Parker-Oldenburg inversion process reiterates the interpretations based on the complete Bouguer anomaly gravity data. The average Moho depth as per the inverted Moho interface is ~ 38 km, the maximum depth (44 km) is seen below areas covered by the Bundelkhand granitoids and the minimum (~32 km) is below the Vindhyan basin sequences outlining the southern Bundelkhand craton. The elongated, shallow Moho topography below the Vindhyan basin suggests that the basin

formation was accompanied with extension of the crust with upwarping of the Moho due to mantle upwelling or emplacement of high-density mantle material at the base of the crust. The high gravity values seen in the gravity map from the inverted Moho interface supports the presence of high-density material at deep crustal depths, possibly between the lower crust and the Moho. This leads to the interpretation that the observed shallow depth to the Moho below the Proterozoic Vindhyan basin is approximately the depth to the top of the underplated material at the base of the crust, above the Moho.

This is further validated by the 2D forward models. The density models constructed for the profiles AA' and BB', illustrate that the central gravity high observed in the complete Bouguer anomaly of the larger study area is due to the presence of a high-density ($3150 \, kg/m^3$) underplating layer above the Moho along with the high-density Bijawar rocks. The underplating gains the maximum thickness (~12 km) below the central portions of the profile AA', showing the large extent of this deep crustal layer within the central parts of the study area. The model computed along the BB' profile shows that the Moho

shallows up under the Deccan trap exposures, indicating uplifted Moho, (~ 37 km from Moho depth map from inversion) because of the extensive volcanic activity that occurred around Cretaceous-Tertiary boundary. Continental rifting by extension, due to the presence of a plume (~2.5−1 Ga) below the Bundelkhand craton is supported by the inferred magma-compensated crustal thinning from the observations and results. The evolution of the marginal Bijawar basin as an intracratonic rift basin, along with the thick sedimentary Vindhyan basin formation, describes the role played by the plume

in the breakup and assembly of the Columbia and Rodinia supercontinents, respectively.

**Data availability**

In this research, open sources free-air gravity anomaly grid data and topography data have been utilized as freely available at the website of the Scripps Institution of Oceanography (https://topex.ucsd.edu/cgi-bin/get_data.cgi). This global gravity

model of 1-minute grids has approximately 2 mGal accuracy and is based on data from the Geosat and ERS-1 satellites, along with new altimeter data from Jason-1 and CryoSat-2 satellites (Smith and Sandwell, 1997; Sandwell et al., 2014; Kende et al., 2017). Thus, all the unprocessed data related to this study can be accessed from https://topex.ucsd.edu/cgi-bin/get_data.cgi

**Author contributions**

APM: Conceptualization, data acquisition, analysis, interpretation, modelling, visualization, and original draft preparation; AM: Conceptualization, formulation of research goals, supervision, resources, data curation, validation, and review and editing.

**Competing interests**

The contact author has declared that none of the authors has any competing interests.

**Funding**

This is a contribution under the project ECR/2015/000247 awarded to the 2nd author from the Science and Engineering Research Board (SERB), Department of Science and Technology, Government of India.

**Acknowledgements**

This is a contribution under the project ECR/2015/000247 awarded to the 2nd author. Authors gratefully acknowledge the
financial support (File No. ECR/2015/000247) of Science and Engineering Research Board (SERB), Department of Science and Technology, Government of India, to carry out this work. The first author thanks the Ministry of Human Resource Development (MHRD), Government of India for awarding the prestigious Prime Minister's Research Fellowship (PMRF).

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

**Table 1: Density values used in the present study, compiled from established literature.**

| Layers | Density (kg/m$^3$) | References |
|---|---|---|
| Recent sediments | 2100 | Prasad et al. (2018) |
| Vindhyan supergroup | 2500 | Mishra (2015); Pal and Kumar (2019) |
| Bijawar basement of Vindhyan | 2840 | Mishra (2015 |
| Bundelkhand granite + basement, Upper crust (average) | 2640 | Podugu et al. (2017); Pati and Singh (2020) |
| Deccan traps | 2850 | Rao et al. (2011) |
| Average Middle and Lower crustal density | 2800 | Rao et al. (2011); Chouhan et al. (2020) |
| Underplated layer | 3150 | Chouhan et al. (2020) |
| Upper mantle | 3300 | Rao et al. (2011); Chouhan et al. (2020) |

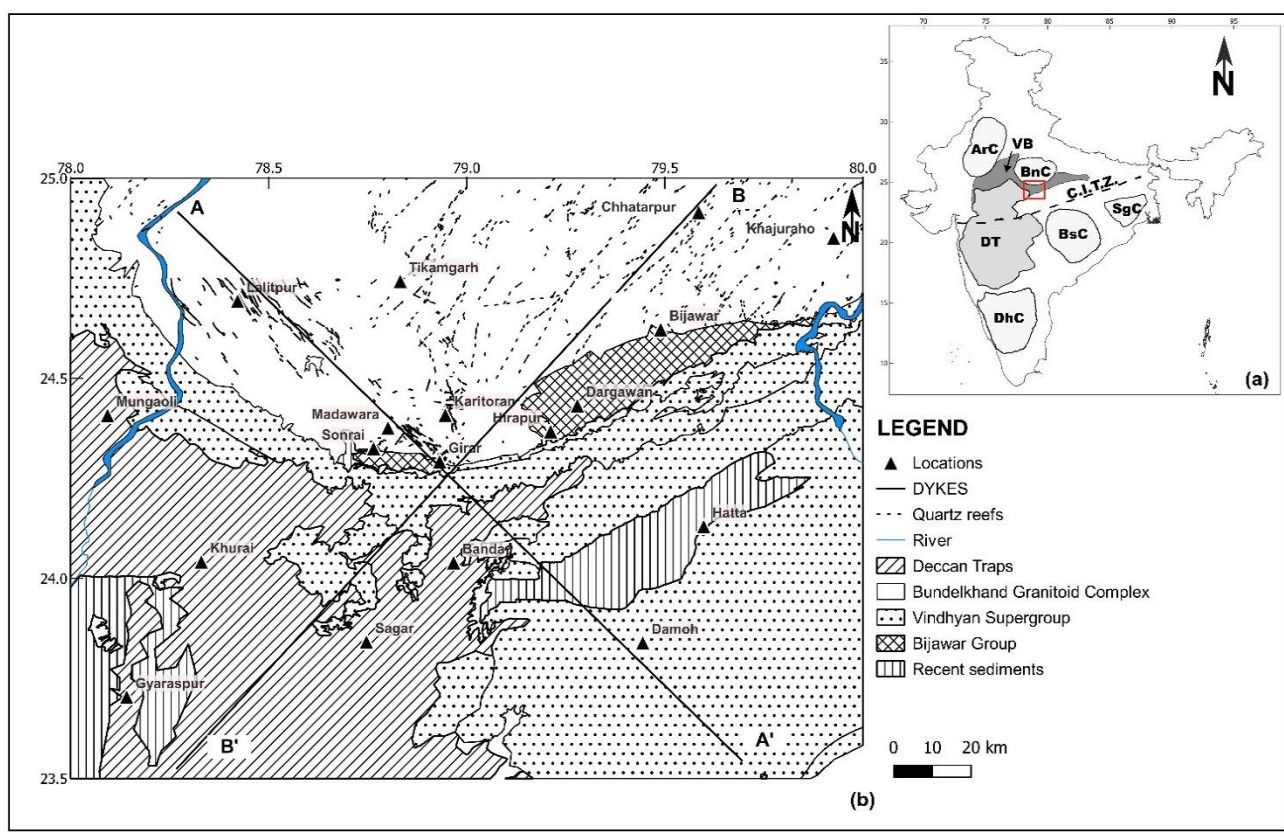

**Figure 1: (a) Position of the Bundelkhand craton and Vindhyan basin with respect to other major cratons of the Indian subcontinent. Bijawar basin forms the base of the Vindhyan basin, and the exposed sequences are shown in Figure 1b. [ArC-**
**Aravalli Craton, BnC- Bundelkhand Craton, VB- Vindhyan Basin, DT- Deccan Traps, DhC- Dharwar Craton, BsC- Bastar Craton, SgC- Singhbhum Craton, C.I.T.Z.- Central Indian Tectonic Zone] (b) General geological setup of the region used for the regional scale study of the craton and surrounding areas along the southern boundary of the craton. The two profiles used for gravity modelling are marked here as AA′ and BB′. (Data source: GSI Bhukosh platform, https://bhukosh.gsi.gov.in/)**

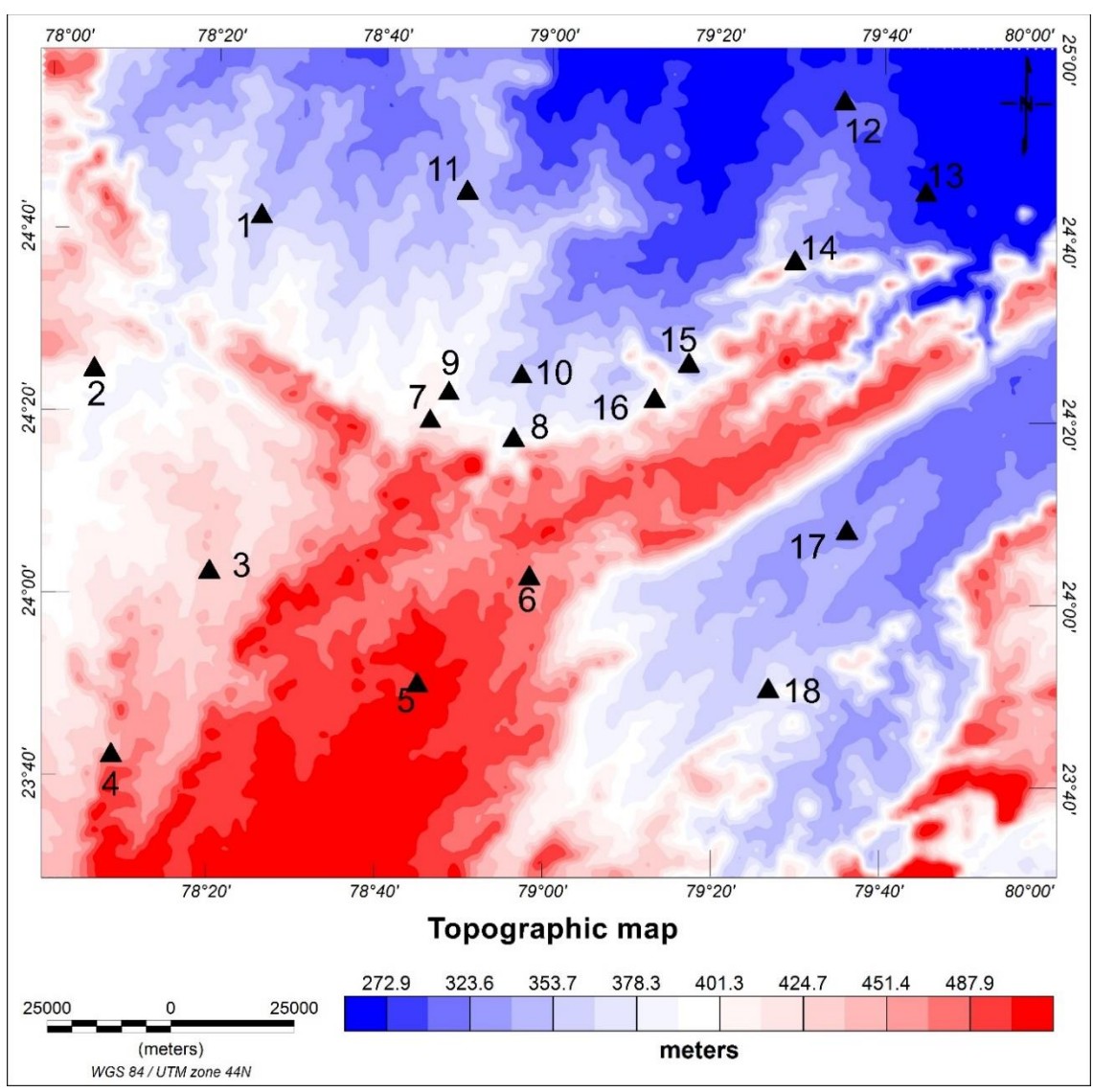

**Figure 2: Topographic map derived from the global 1-minute topography grids available on the website of the Scripps Institution of Oceanography, (https://topex.ucsd.edu/WWW_html/mar_topo.html;** https://topex.ucsd.edu/cgi-bin/get_data.cgi). **Locations: (1)Lalitpur, (2)Mungaoli, (3)Khurai, (4)Gyaraspur, (5)Sagar, (6)Banda, (7)Sonrai, (8)Girar, (9)Madawara, (10)Karitoran, (11)Tikamgarh, (12)Chhatarpur, (13)Khajuraho (14)Bijawar, (15)Dargawan, (16)Hirapur, (17)Hatta, (18)Damoh.**



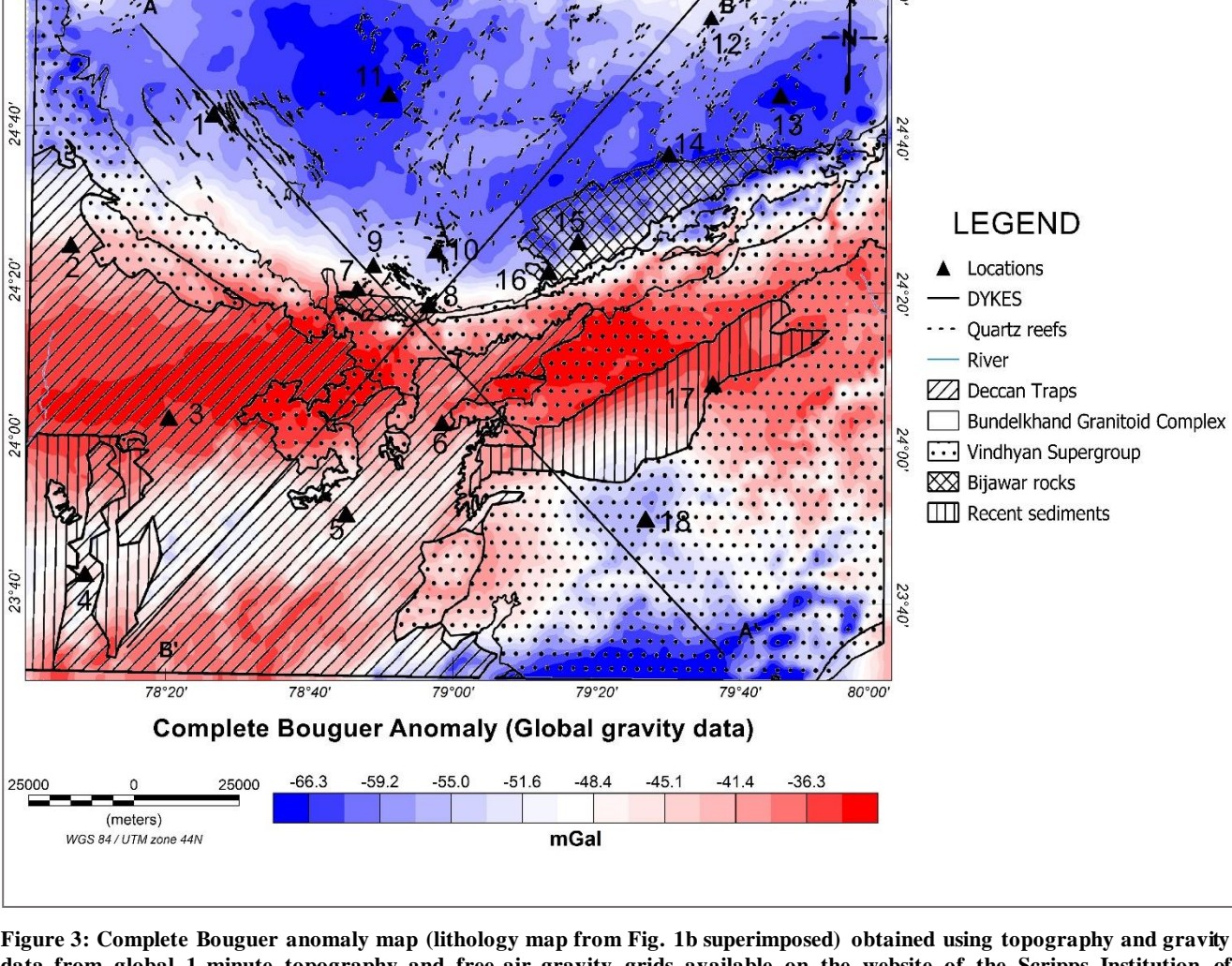

**Figure 3: Complete Bouguer anomaly map (lithology map from Fig. 1b superimposed) obtained using topography and gravity data from global 1-minute topography and free-air gravity grids available on the website of the Scripps Institution of Oceanography, (https://topex.ucsd.edu/WWW_html/mar_topo.html; https://topex.ucsd.edu/cgi-bin/get_data.cgi). Locations: (1)Lalitpur, (2)Mungaoli, (3)Khurai, (4)Gyaraspur, (5)Sagar, (6)Banda, (7)Sonrai, (8)Girar, (9)Madawara, (10)Karitoran, (11)Tikamgarh, (12)Chhatarpur, (13)Khajuraho, (14)Bijawar, (15)Dargawan, (16)Hirapur, (17)Hatta, (18)Damoh.**


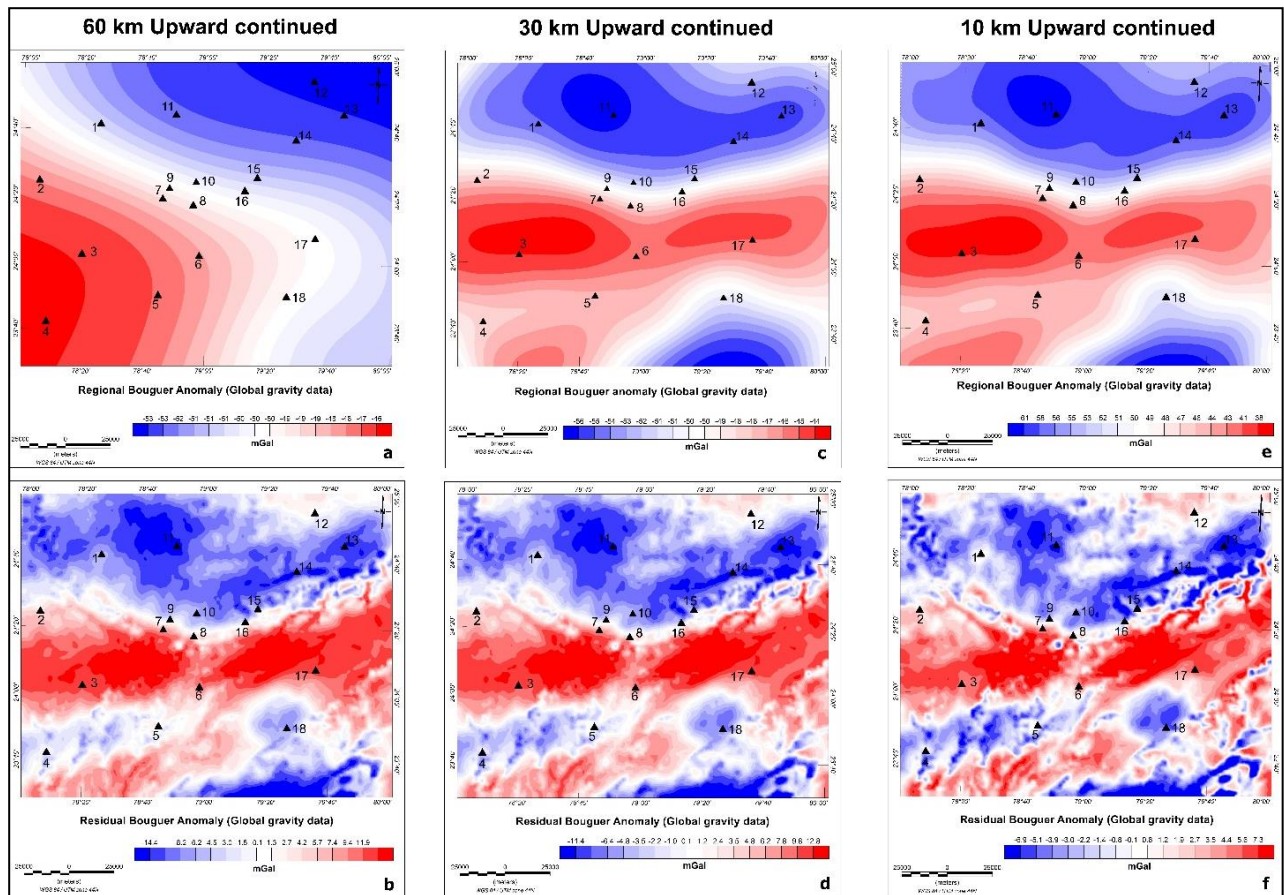

**Figure 4: (a)** Regional gravity anomaly map of the global gravity data, upward continued up to 60 km, **(b)** Residual gravity anomaly map of the global grid data, obtained after subtracting the 60 km upward continued regional gravity anomaly from complete Bouguer anomaly. **(c)** Regional gravity anomaly map of the global gravity data, upward continued up to 30 km, **(d)** Residual gravity anomaly map of the global grid data, obtained after subtracting the 30 km upward continued regional gravity anomaly from complete Bouguer anomaly. **(e)** Regional gravity anomaly map of the global gravity data, upward continued up to 10 km, **(f)** Residual gravity anomaly map of the global grid data, obtained after subtracting the 10 km upward continued regional gravity anomaly from complete Bouguer anomaly. Locations: (1)Lalitpur, (2)Mungaoli, (3)Khurai, (4)Gyaraspur, (5)Sagar, (6)Banda, (7)Sonrai, (8)Girar, (9)Madawara, (10)Karitoran, (11)Tikamgarh, (12)Chhatarpur, (13)Khajuraho, (14)Bijawar, (15)Dargawan, (16)Hirapur, (17)Hatta, (18)Damoh.

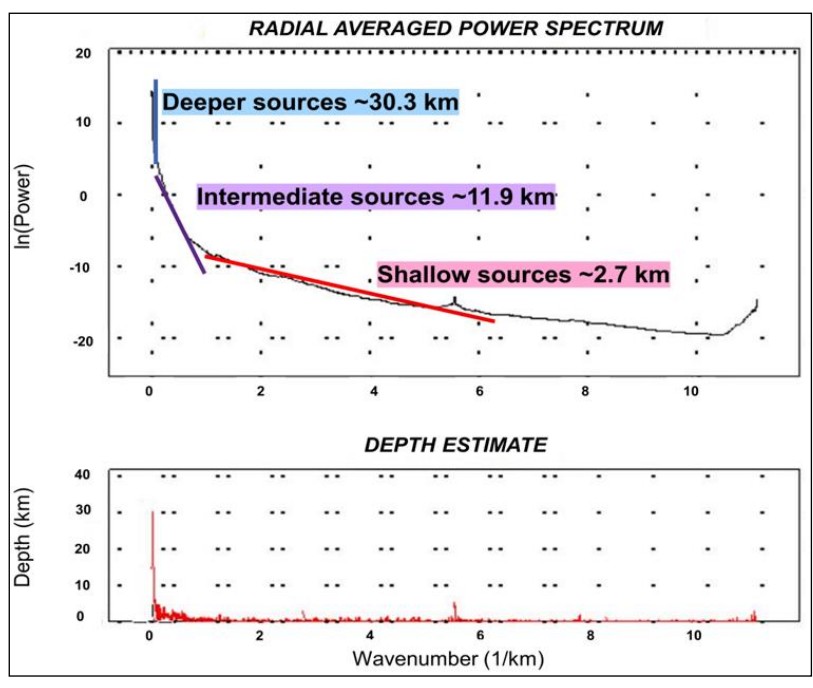

**Figure 5: Radially averaged power spectrum plot (upper panel) with the corresponding depth estimates plot (lower panel) for the complete Bouguer anomaly data (Fig. 3).**

### Topography of inverted Moho interface obtained from Bouguer gravity data

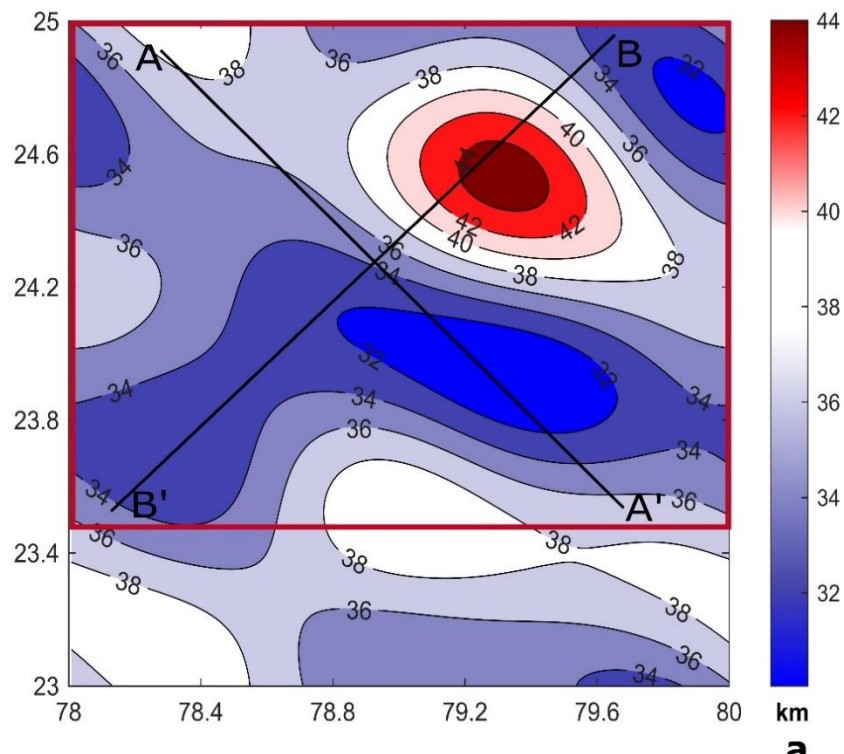

**a**

### Gravity anomaly map obtained from inverted Moho topography

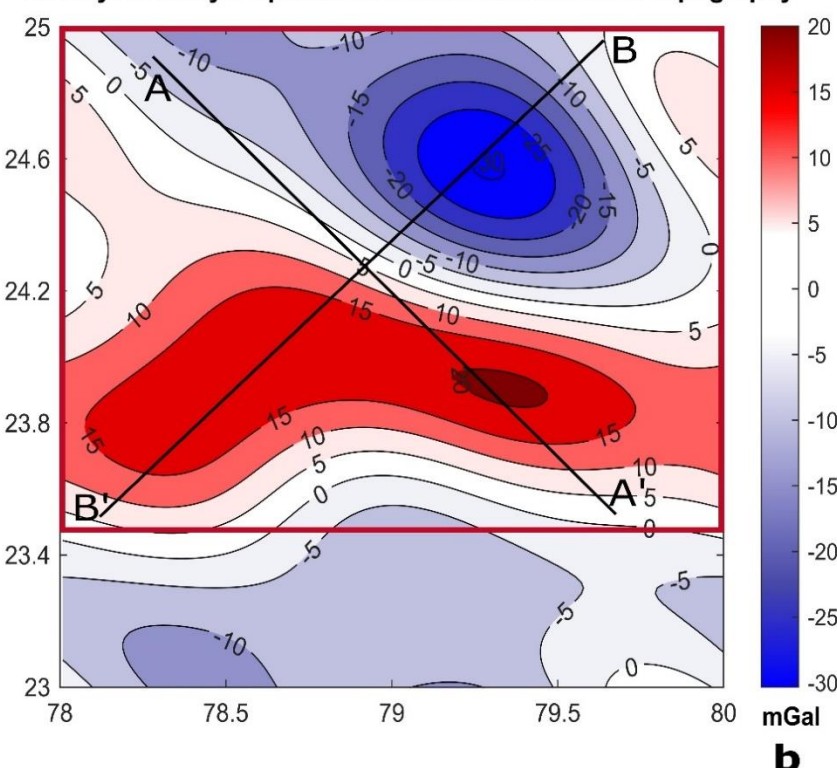

**b**

**Figure 6:** (a) **Moho topography map obtained by applying Parker-Oldenburg method on the complete Bouguer anomaly data of Fig. 3. Contour interval is 2 km, (b) Gravity map obtained using the inverted Moho depths from Fig. 6a. The contour interval is 5 km. The red box marks the study area.**

# Profile AA'

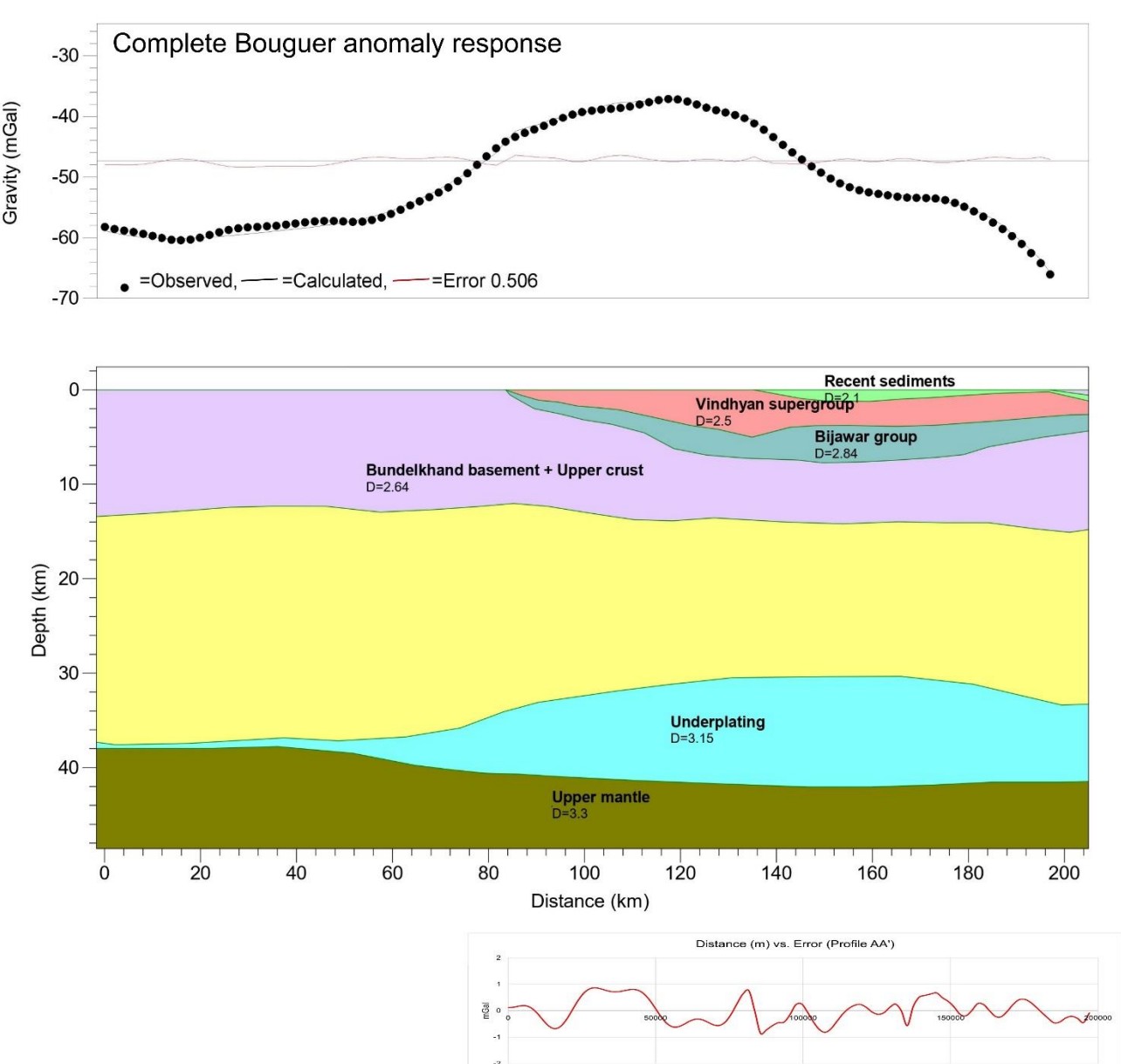

**Figure 7: Observed complete Bouguer anomaly data (Fig. 3) and calculated Bouguer gravity responses (upper panel) with the computed density models (lower panel) along the AA' profile (Fig. 1b). Corresponding distance vs misfit error along profile A A′ is shown below as a separate panel.**

# Profile BB'

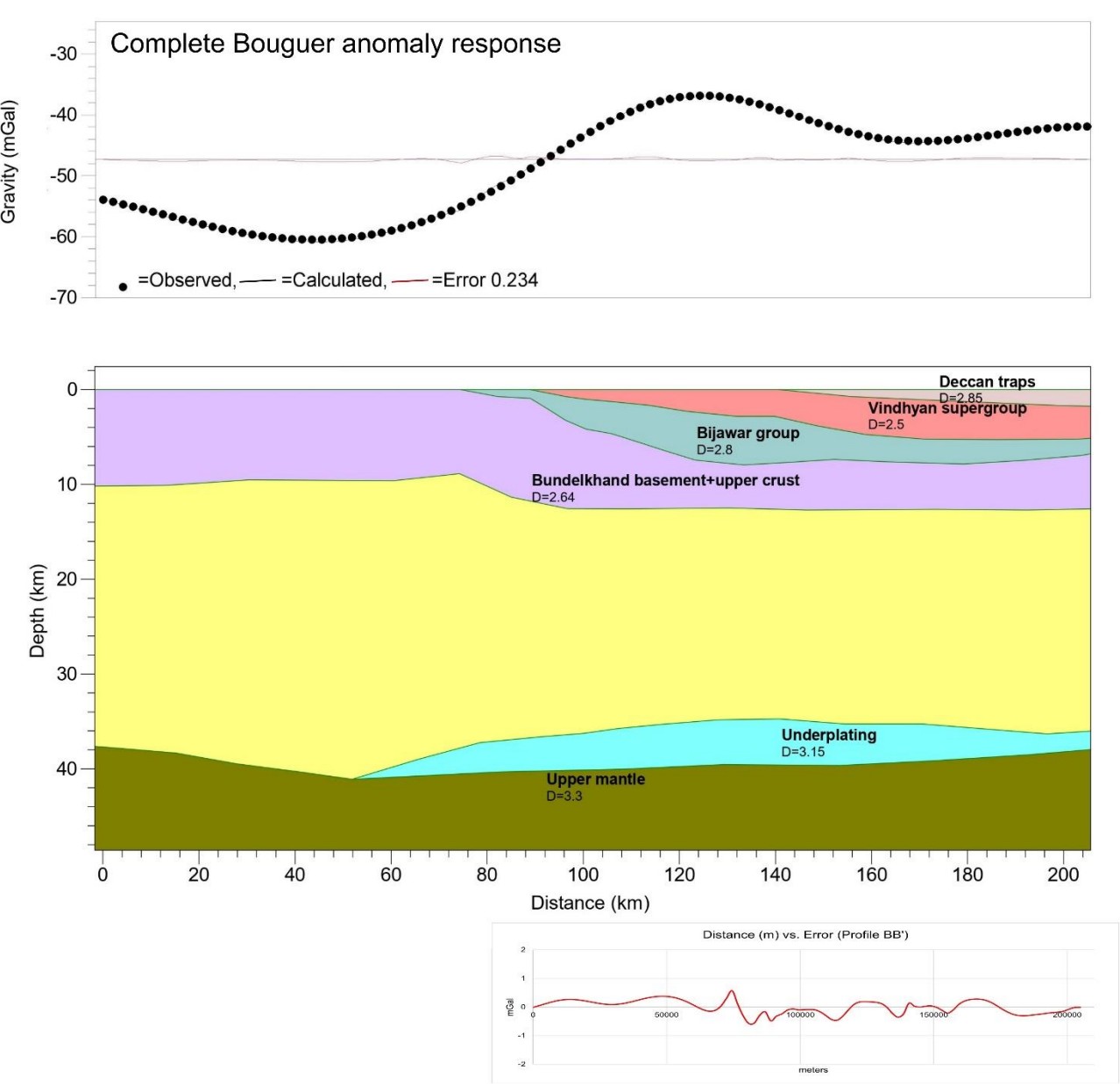

**Figure 8: Observed complete Bouguer anomaly data (Fig. 3) and calculated Bouguer gravity responses (upper panel) with the computed density models (lower panel) along the BB′ profile (Fig. 1b). Corresponding distance vs misfit error along profile BB′ is shown below as a separate panel.**

770

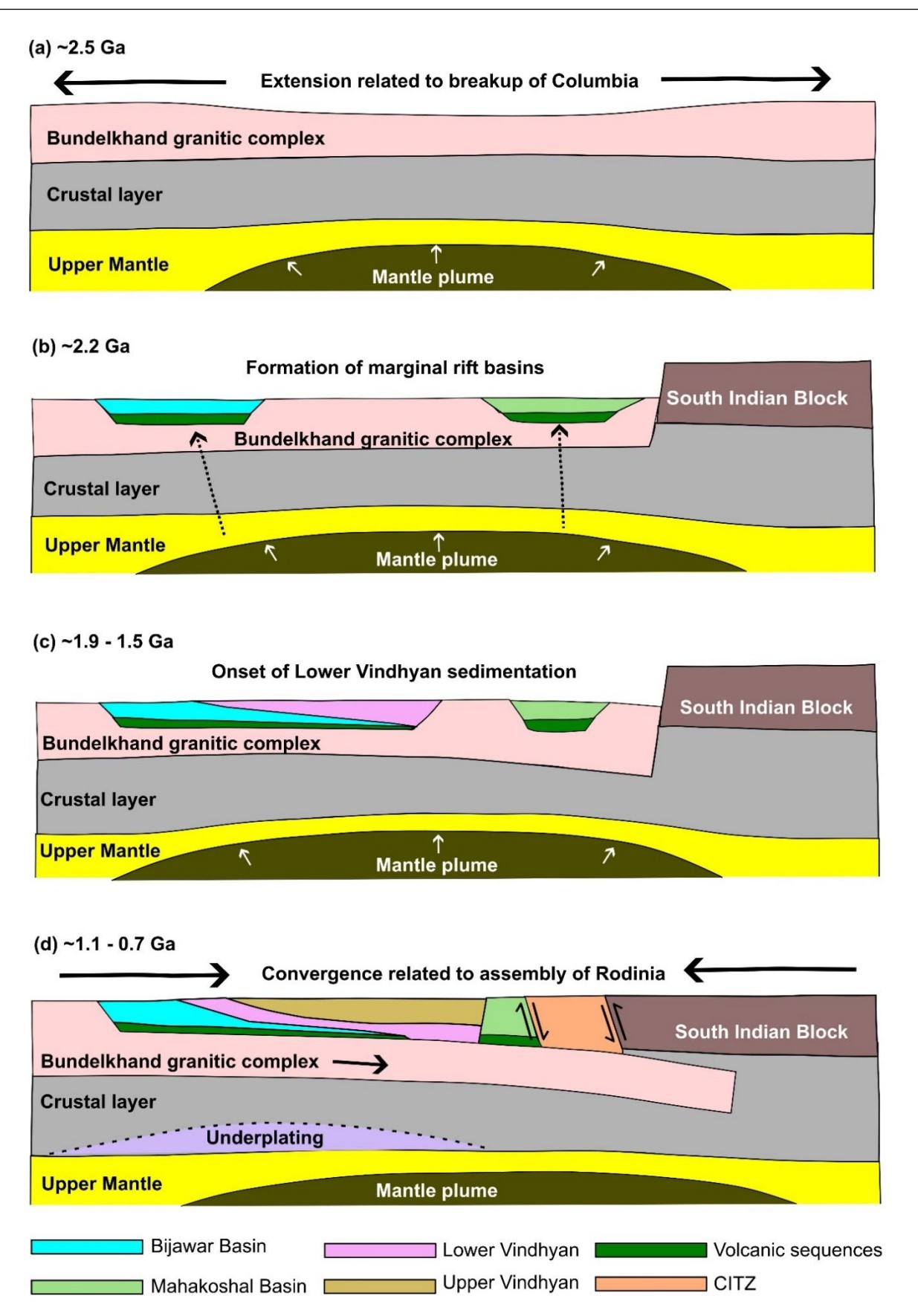

**Figure 9: Schematic representation of the sequence of tectonic evolution of Bijawar and Vindhyan basin due to the presence of a plume below the Bundelkhand craton up to the formation of the CITZ. Modified using the works of Mishra (2015), Patranabis-Deb and Saha (2020), Colleps et al. (2021).**

775