# Peer review of "Magmatic underplating associated with Proterozoic basin formation: insights from gravity study over the southern margin of Bundelkhand craton, India"

_EGUsphere, 2023_

## Referee Comment (RC2)

**egusphere-2023-1389 Manuscript review**

In the manuscript "Magmatic underplating with Proterozoic basin formation: insights from gravity study over the southern margin of Bundelkhand craton, India", the authors investigate the tectonic mechanisms leading to the formation of two basins: the Bijawar and Vindhyan basins, located in proximity of the southern boundary of the Bundelkhand craton.
The authors suggest an extensional mechanism favored by magmatic underplating below the crust at relatively shallow depth (ca. 30 km), based on the analysis and modeling of global gravity data, and Bouguer gravity anomaly in particular.
The scientific question concerning the origin of the basins is relevant and of interest for the scientific community, and this is well explained in the manuscript Introduction. In general, the chosen methods are suitable for the proposed investigation and the manuscript is well written.

However, I think that some methodological choices require better clarification and support, to strengthen the results interpretation. Also, the quality of some of the figures requires improvement.

Here I provide some important points, which I believe should be addressed prior to considering the manuscript for final publication:

1) The authors use global gravity data and upward continue Bouguer gravity anomaly to investigate the crustal structure within the study area. I understand that upward continuing acts as a low pass filter to enhance the long-wavelength regional Bouguer anomaly trend.

   Please, can the authors explain why they chose 30 km elevation for the upward continuation? Is 30 km the elevation at which the effect of surface structures become negligible?

   It would be interesting to provide a few test examples of upward continuation at different elevations (possibly as supplementary material); also, see e.g. Zeng et al. (2007), *Geophysics*, for approaches to estimate ideal elevation for upward continuation.

   Also, the terrain correction seems to be small (less than 1 mGal), but topography ranges from 175 m to 617 m (line 152). Can the authors provide a topographic map of the study area, together with the grid points from the global gravity data they used?

2) The authors extract the regional Bouguer anomaly trend and separate it from the gravity effect of surface geological structures (residuals). However, they often refer to "Bouguer gravity anomaly" throughout the manuscript.
   Please, state explicitly which type of gravity data product is used for RAPS analysis, 3D Moho depth inversion, and 2D forward gravity modeling respectively.

   Is the upward-continued regional Bouguer trend used in every analysis? If yes, can it constrain the shallower (hence smaller wavelength) structures in the 2D forward

modeling (e.g. basin structures at a few km depth below the surface in Figures 6 and 7)?

3) Could the authors spend a few more words on the RAPS technique? The output provides a deep density contrast interface at ca. 30.3 km depth. What is the resolution expected from this technique, which, if I understand well, provides a 1D average information across the study area?

4) The authors perform a 3D inversion for the Moho topography. The algorithm they use requires assuming a mean depth $z_0$ and a density contrast. Why did they use 36 km for $z_0$? And not e.g. 30 km as obtained from the previous RAPS analysis?

   And related to this, how do these results change as a function of $z_0$ and density contrast? Please, provide a sensitivity test for these parameters, and a resolution test for the inversion algorithm application.

5) I believe a few important points should be clarified in the 2D modeling stage.
   The authors perform 2D forward modeling based on the formulas by Talwani et al. (1959). These formulas are best suited when both the data, and the target area, present a 2.5D symmetry (e.g. changing properties along the x-z plane, and no changes along y-axis); see e.g. Scarponi et al. (2021), *Frontiers* for one example.
   In particular, profile AA' seems not to be perpendicular to visible 2.5D structures, nor in the gravity data or in the underlying, inverted Moho structure (based on Figure 2 and Figure 5a). A slight-to-moderate rotation of profile AA' around its center (e.g. +-20 degrees) could potentially provide a different gravity data profile, and hence lead to different results and interpretation.

   a) The authors could consider using a 3D inversion software (see e.g. IGMAS+ in Spooner et al. 2019, *Solid Earth*). If not 3D, how would the 2D gravity profiles (data and models) look like along a set of parallel profiles (e.g. at constant longitude)? Would the results, and hence interpretation, change along a different profile than AA'? This should be tested and discussed before interpretation.

   b) Paragraph 3.5 on the construction of the 2D profiles never mentions incorporating the results from the 3D Moho depth inversion. Were these Moho results neglected in the creation of the 2D models shown in Figure 6 and 7? If yes, why?
   The authors mention the RAPS estimate as reference used in the profiles, but RAPS provides an inherently 1D average information. Moreover, profile BB' shows no interfaces around 30 km: please, can the authors explain the reason for this?

   According to Figure 5a, the computed Moho depths obtained along BB' range from 44 km to 34 km depth, but this seems not to be the case when looking at Figure 7. This should be clarified (partially applies also to profile AA' and Figure 6).

   c) In the definition of the structures within profiles AA' and BB', the authors refer to a list of previous investigations, to be used as external constraints. This is OK in principle. However, these external constraints are not explicitly indicated in Figure 6 and 7.

Which geometries were imported as unmodified external information? Which ones were generated and/or modified by the authors? This information is not clear and should be made explicit. The authors could also show in Figures 6 and 7, how their new Moho estimate compares to the external information they refer to.

Clarifying the points above is important to discuss the fit to the gravity data along the selected profiles. For example, was the geometry of the underplating structure in Figure 7 imported from external sources? How would the forward modeling compare to the data, with a different, or without, the underplating layer along AA'?

To address these points, I would advise starting by: 1) apply RAPS; 2) use the RAPS deepest interface estimate as z0 for the Moho inversion (provide sensitivity and resolution tests); 3) use the obtained Moho depth as starting geometrical constraint, together with those existing in the literature, either for 3D modeling, or for a set of 2D profiles (as much as possible along structures with 2.5D symmetry), providing support for the chosen 2D profiles; 4) test if the deeper underplating layers can be resolved by gravity along the chosen profiles.

Here I provide few additional specific comments:

-   Figure 1b is not readable and should be larger. Figure 1a is readable, but please consider using different colors to highlight the different geological units.
    The perimeter box of figure 1a should appear in figure 1b to show its location;

-   Figure 5 should at least contain a residual map (synthetics minus observations).
    It would be also beneficial to add a plot for RMS versus iteration number, to show the RMS reduction during the inversion, and a sensitivity and resolution tests (possibly in a different Figure);

-   Figure 6 and 7 should show explicitly which geometries were imported as unmodified external constraints for the construction of the models. They should also show the Moho depth as obtained from the 3D inversion (Figure 5a).
    Also, the top banner in Figures 6 and 7 is not very clear: is it gravity or Bouguer anomaly? Please, plot the error on a separate independent scale to be more readable

-   line 175: by "Bouguer anomaly" you mean the upward-continued regional trend? Please, specify. Same for line 219, 239, 270, 286, 295 and so on.

-   Line 244: Does GMSYS perform 2D forward modeling or 2D inversion?

    This is a crucial detail. If it performs inversion, then more information is needed here. Or, have you tested several candidate profiles? Please, explain.

-   Line 375-377: To my understanding, you obtain a Moho from 3D gravity inversion. But you do NOT obtain a Moho depth from 2D forward gravity modeling (see also question above). If you do not perform 2D inversion, then 2D forward modeling can only validate a certain profile, but not "provide" or "obtain" from it.

This is better formulated later in the conclusions, at line 442 "[...] validated by the 2D [...]". Please, This should be clarified. And finally, why not using the 3D Moho results in the construction of the 2D models?

---

## Author Response (AR1)

**RESPONSE TO THE REFEREES**

We, the authors, would like to extend our sincere gratitude to the reviewers for their valuable time and efforts invested in reading the manuscript. We appreciate and acknowledge all the comments, feedback, and constructive suggestions provided for the further improvement of the manuscript by the reviewers. We have responded to each of the comments with the best possible clarifications, as well as considering the feedback, we have carefully incorporated most of the suggestions of the reviewers in the revised manuscript. The responses to each of the specific comments and the corresponding figures for the clarifications are compiled below.

**A. Comments from Referee #1**

GENERAL COMMENT

*The manuscript attempts to relate the creation and evolution of the proterozoic Bijawar and Vindhyan sedimentary basins of India to the plume-related rifting mechanism and the associated mafic subcrustal underplate emplaced at a depth of greater than 30 km. The authors used the global gravity grid data to show the spatial and depth extent of the high density mafic underplate in the two forward models they prepared. The topic is important and interesting and the manuscript is well written. However, there are points that need addressing in order to improve the quality of the manuscript and to make it relevant and make it suitable for publication in the form of a research paper.*

SPECIFIC COMMENTS

1. *The title: Basin formation and evolution is attributed to the combined effects of all the sublithospheric actions including underplating as well as the lithospheric plate actions and the supergene action of the lithosphere itself as expressed in all the processes that take place on the earth's surface. The title, as it is and at a glance, seems to carry the idea that underplating alone can play a significant role in basin formation.*

2. *The authors have not shown the locations of the two basins. What they show on the map is the lithostratigraphic units bearing the names of the two basins.*

3. *What is the rationale for upward continuing the gravity data to a 30 km height? It is more reasonable to base the continuation height on the corresponding depth estimates obtained from the radially averaged power spectrum.*

4. *Gravity modeling is loosely constrained with only limited information. Hence the modeling result should be interpreted with caution.*

5. *To generalize that the high density anomaly has sources extending from deep to shallow is an over simplification. The authors have done upward continuation to a single height and they base their conclusion on it. Try to upward continue to a height of 60 (corresponding to sources at a depth of 30 km and below) to justify presence of underplate at depths in the order of 30 km in the models. In addition, the central part of the models (central region) do not show the extension of high anomaly sources to shallower depths as shown by the residual anomaly.*

6. *On Line 332 the paper asserts that there is striking similarity between the inverted Moho topography and the gravity signature. Please tone down this assertion. There is similarity but pay attention to the following remarks: a. the gravity anomaly from the inverted Moho topography shows a slightly southward shift and centered at the southern margin of the regional anomaly. b. at the*

*southwestern corner the effect is in fact opposite. There is high anomaly in the both the regional and residual gravity but low anomaly in the gravity obtained from the Moho.*

*7.        I have serious reservation on the contradictory results obtained from the two approaches of the gravity modeling. The authors obtained the undulation of the Moho interface using the downward continuation formula as given by the modification of Parker-Oldenburg algorithm (Fig. 5a). As is also noted by the authors the Moho depth undulations obtained by this method beneath the basins where underplate is observed do not correspond to the depths obtained by forward modeling (Fig. 6 & 7). The underplate top is considered as Moho in Fig. 5a whereas the underplate bottom is considered as Moho in both Fig. 6 & 7. In addition, there is also a clear discrepancy in areas where there is no underplate. Compare, for example, Fig. 5a with Fig. 7. In northeast where there is no underplate the Moho depth varies between 37 and 39 km in the forward model in Fig. 7 whereas in Fig. 5a the depth variation for the same area is between 33 and 44 km. These contradictions should not have occurred since there is a clear density contrast between the mantle and the underplate and the Parker-Oldenburg approach is capable of recognizing the difference between them.*

TECHNICAL CORRECTIONS

*1.        On Line 29 "…..attributed to the presence of underplated layers, like rift basins…" Please remove the word "like".*

*2.        On Lines 194 and 197- 8 the phrase h(x) and zo are redundant. Please eliminate the repetition.*

*3.        On Line 346 it must be "crustal uplift" and not "crustal upliftment".*

*4.        In Fig. 5   contour values can be indicated on every other line to avoid congestion.*

**B.  Comments from Referee #2**

*In the manuscript "Magmatic underplating with Proterozoic basin formation: insights from gravity study over the southern margin of Bundelkhand craton, India", the authors investigate the tectonic mechanisms leading to the formation of two basins: the Bijawar and Vindhyan basins, located in proximity of the southern boundary of the Bundelkhand craton. The authors suggest an extensional mechanism favored by magmatic underplating below the crust at relatively shallow depth (ca. 30 km), based on the analysis and modeling of global gravity data, and Bouguer gravity anomaly in particular.*

*The scientific question concerning the origin of the basins is relevant and of interest for the scientific community, and this is well explained in the manuscript Introduction. In general, the chosen methods are suitable for the proposed investigation and the manuscript is well written.*

*However, I think that some methodological choices require better clarification and support, to strengthen the results interpretation. Also, the quality of some of the figures requires improvement.*

*Here I provide some important points, which I believe should be addressed prior to considering the manuscript for final publication:*

*1)        The authors use global gravity data and upward continue Bouguer gravity anomaly to investigate the crustal structure within the study area. I understand that upward continuing acts as a low pass filter to enhance the long-wavelength regional Bouguer anomaly trend.*

*Please, can the authors explain why they chose 30 km elevation for the upward continuation? Is 30 km the elevation at which the effect of surface structures become negligible?*

*It would be interesting to provide a few test examples of upward continuation at different elevations (possibly as supplementary material); also, see e.g. Zeng et al. (2007), Geophysics, for approaches to estimate ideal elevation for upward continuation.*

*Also, the terrain correction seems to be small (less than 1 mGal), but topography ranges from 175 m to 617 m (line 152). Can the authors provide a topographic map of the study area, together with the grid points from the global gravity data they used?*

*2)     The authors extract the regional Bouguer anomaly trend and separate it from the gravity effect of surface geological structures (residuals). However, they often refer to "Bouguer gravity anomaly" throughout the manuscript.*

*Please, state explicitly which type of gravity data product is used for RAPS analysis, 3D Moho depth inversion, and 2D forward gravity modeling respectively.*

*Is the upward-continued regional Bouguer trend used in every analysis? If yes, can it constrain the shallower (hence smaller wavelength) structures in the 2D forward modeling (e.g. basin structures at a few km depth below the surface in Figures 6 and 7)?*

*3)     Could the authors spend a few more words on the RAPS technique? The output provides a deep density contrast interface at ca. 30.3 km depth. What is the resolution expected from this technique, which, if I understand well, provides a 1D average information across the study area?*

*4)     The authors perform a 3D inversion for the Moho topography. The algorithm they use requires assuming a mean depth z0 and a density contrast. Why did they use 36 km for z0? And not e.g. 30 km as obtained from the previous RAPS analysis?*

*And related to this, how do these results change as a function of z0 and density contrast? Please, provide a sensitivity test for these parameters, and a resolution test for the inversion algorithm application.*

*5)     I believe a few important points should be clarified in the 2D modeling stage.*

*The authors perform 2D forward modeling based on the formulas by Talwani et al. (1959). These formulas are best suited when both the data, and the target area, present a 2.5D symmetry (e.g. changing properties along the x-z plane, and no changes along y-axis); see e.g. Scarponi et al. (2021), Frontiers for one example. In particular, profile AA' seems not to be perpendicular to visible 2.5D structures, nor in the gravity data or in the underlying, inverted Moho structure (based on Figure 2 and Figure 5a). A slight-to-moderate rotation of profile AA' around its center (e.g. +20 degrees) could potentially provide a different gravity data profile, and hence lead to different results and interpretation.*

*a)     The authors could consider using a 3D inversion software (see e.g. IGMAS+ in Spooner et al. 2019, Solid Earth). If not 3D, how would the 2D gravity profiles (data and models) look like along a set of parallel profiles (e.g. at constant longitude)? Would the results, and hence interpretation, change along a different profile than AA'? This should be tested and discussed before interpretation.*

b)      Paragraph 3.5 on the construction of the 2D profiles never mentions incorporating the results from the 3D Moho depth inversion. Were these Moho results neglected in the creation of the 2D models shown in Figure 6 and 7? If yes, why?

The authors mention the RAPS estimate as reference used in the profiles, but RAPS provides an inherently 1D average information. Moreover, profile BB' shows no interfaces around 30 km: please, can the authors explain the reason for this?

According to Figure 5a, the computed Moho depths obtained along BB' range from 44 km to 34 km depth, but this seems not to be the case when looking at Figure 7. This should be clarified (partially applies also to profile AA' and Figure 6).

c)      In the definition of the structures within profiles AA' and BB', the authors refer to a list of previous investigations, to be used as external constraints. This is OK in principle. However, these external constraints are not explicitly indicated in Figure 6 and 7.

Which geometries were imported as unmodified external information? Which ones were generated and/or modified by the authors? This information is not clear and should be made explicit. The authors could also show in Figures 6 and 7, how their new Moho estimate compares to the external information they refer to.

Clarifying the points above is important to discuss the fit to the gravity data along the selected profiles. For example, was the geometry of the underplating structure in Figure 7 imported from external sources? How would the forward modeling compare to the data, with a different, or without, the underplating layer along AA'?

To address these points, I would advise starting by: 1) apply RAPS; 2) use the RAPS deepest interface estimate as z0 for the Moho inversion (provide sensitivity and resolution tests); 3) use the obtained Moho depth as starting geometrical constraint, together with those existing in the literature, either for 3D modeling, or for a set of 2D profiles (as much as possible along structures with 2.5D symmetry), providing support for the chosen 2D profiles;

4) test if the deeper underplating layers can be resolved by gravity along the chosen profiles.

Here I provide few additional specific comments:

-       Figure 1b is not readable and should be larger. Figure 1a is readable, but please consider using different colors to highlight the different geological units.

The perimeter box of figure 1a should appear in figure 1b to show its location;

-       Figure 5 should at least contain a residual map (synthetics minus observations).

It would be also beneficial to add a plot for RMS versus iteration number, to show the RMS reduction during the inversion, and a sensitivity and resolution tests (possibly in a different Figure);

-       Figure 6 and 7 should show explicitly which geometries were imported as unmodified external constraints for the construction of the models. They should also show the Moho depth as obtained from the 3D inversion (Figure 5a).

Also, the top banner in Figures 6 and 7 is not very clear: is it gravity or Bouguer anomaly? Please, plot the error on a separate independent scale to be more readable

- line 175: by "Bouguer anomaly" you mean the upward-continued regional trend? Please, specify. Same for line 219, 239, 270, 286, 295 and so on.

- Line 244: Does GMSYS perform 2D forward modeling or 2D inversion?

This is a crucial detail. If it performs inversion, then more information is needed here. Or, have you tested several candidate profiles? Please, explain.

- Line 375-377: To my understanding, you obtain a Moho from 3D gravity inversion. But you do NOT obtain a Moho depth from 2D forward gravity modeling (see also question above). If you do not perform 2D inversion, then 2D forward modeling can only validate a certain profile, but not "provide" or "obtain" from it.

This is better formulated later in the conclusions, at line 442 "[...] validated by the 2D [...]". Please, This should be clarified. And finally, why not using the 3D Moho results in the construction of the 2D models?

**C. Authors' response to the comments from Referee #1**

**Note:** Figures numbers are given as per the sequence of appearances of the figures in the responses to the comments of reviewer # 1; wherever required corresponding figure number in the revised manuscript is also referred.

SPECIFIC COMMENTS

*1. The title: Basin formation and evolution is attributed to the combined effects of all the sublithospheric actions including underplating as well as the lithospheric plate actions and the supergene action of the lithosphere itself as expressed in all the processes that take place on the earth's surface. The title, as it is and at a glance, seems to carry the idea that underplating alone can play a significant role in basin formation.*

**Response:** We agree with the reviewer, and we want to clarify here that the title used for this manuscript indicates that the magmatic underplating that has been inferred in this study is a piece of evidence that can be associated with the rift basin formation. However, we have not mentioned it as the sole mechanism for the rift formation.

*2. The authors have not shown the locations of the two basins. What they show on the map is the lithostratigraphic units bearing the names of the two basins.*

**Response:** Thank you for highlighting the issue. Accordingly, Figure 1 has been modified and the location of the Vindhyan basin is now shown in Figure 1a. Rocks belonging to the Bijawar basin now form the base of the Vindhyan basin (Basu and Bickford, 2015; Mishra, 2015) and only parts of its rock sequences are exposed along the southern margin of the Bundelkhand craton (Fig. 1b). Hence, the exposures of the Bijawar supergroup, that belong to the respective basin are now shown in the geological map of Figure 1a-b (below as well as in the revised manuscript).

[Figure]

**Figure 1: (a) Position of the Bundelkhand craton and Vindhyan basin with respect to other major cratons of the Indian subcontinent. Bijawar basin forms the base of the Vindhyan basin and the exposed sequences are shown in Figure 1b. (b) General geological setup of the region used for the regional scale study of the craton and surrounding areas along the southern boundary of the craton. The two profiles used for gravity modelling are marked here as AA′ and BB′.**

*3. What is the rationale for upward continuing the gravity data to a 30 km height? It is more reasonable to base the continuation height on the corresponding depth estimates obtained from the radially averaged power spectrum.*

**Response:** The choice of the upward continuing heights was based on a trial-and-error approach as suggested by Gupta and Ramani (1980). Based on this, the 30 km upward continued regional anomaly showed some similarities with the overall trend observed in the Bouguer anomaly map (Fig. 2, below; and Fig. 3 of revised manuscript). Corresponding residual anomaly obtained after removing the 30 km upward continued regional anomaly also showed some correlations with the lithological units observed in Figure 1b. As a result, these maps were only included in the manuscript. We have utilized these results to qualitatively understand the continuation of the high-density sources at different depth levels.

We agree with the reviewer's point that it is theoretically more reasonable to consider the upward continuation height as twice the value of the source depth (Jacobsen, 1987; Meng et al., 2009; Pal and Kumar, 2019; Kebede et al., 2020). Thus, following the reviewer's suggestion, we have now included the results based on the upward continuation heights of 60 km, 30 km, and 10 km (Figs. 3A, 3B, 3C, respectively, see below; and Fig. 4a-f of revised manuscript) corresponding to the depth estimates from radially averaged power spectrum analysis, i.e., ~30.3 km, ~11.9 km, and ~2.7 km, respectively. The 60 km, 30 km, and 10 km upward continued regional anomalies, all showing highs occurring in the SW corner (Figs. 3A(a), 3B(a), and 3C(a), respectively). The regional and residual anomaly maps obtained by the 10 km upward continuation method (Fig. 3C) show similarity to those obtained from the 30 km upward continuation method (Fig. 3B). These suggest the continuation of high-density sources from deeper to shallower depths. Based on the geological setup of this region and the above observations, it can be inferred that the upwelling magma and eventual magmatic emplacement as an underplated layer at the lower crustal levels as well as the volcanogenic rock sequences of the Bijawar group at shallower depth (Mishra, 2015) may have caused such anomaly pattern.

[Figure]

**Figure 2: Complete Bouguer anomaly map (lithology map from Fig. 1 superimposed) obtained using topography and gravity data from global 1-minute topography and free-air gravity grids available on the website of the Scripps Institution of Oceanography, (https://topex.ucsd.edu/WWW_html/mar_topo.html; https://topex.ucsd.edu/cgi-bin/get_data.cgi). Locations: (1)Lalitpur, (2)Mungaoli, (3)Khurai, (4)Gyaraspur, (5)Sagar, (6)Banda, (7)Sonrai, (8)Girar, (9)Madawara, (10)Karitoran, (11)Tikamgarh, (12)Chhatarpur, (13)Khajuraho, (14)Bijawar, (15)Dargawan, (16)Hirapur, (17)Hatta, (18)Damoh.**

[Figure]

**Figure 3: A. (a) Regional Bouguer anomaly map of the global gravity data, upward continued up to 60 km, (b) Residual gravity anomaly map of the global grid data, obtained after subtracting the 60 km upward continued regional gravity anomaly from complete Bouguer anomaly. B. (a) Regional gravity anomaly map of the global gravity data, upward continued up to 30 km, (b) Residual gravity anomaly map of the global grid data, obtained after subtracting the 30 km upward continued regional gravity anomaly from complete Bouguer anomaly. C. (a) Regional gravity anomaly map of the global gravity data, upward continued up to 10 km, (b) Residual gravity anomaly map of the global grid data, obtained after subtracting the 10 km upward continued regional gravity anomaly from complete Bouguer anomaly.**

*4.  Gravity modelling is loosely constrained with only limited information. Hence the modeling result should be interpreted with caution.*

**Response:** We agree with the reviewer's observation that gravity modelling is constrained with only limited information. This is due to the non-availability of adequate studies over the present study area along the southern margin of Bundelkhand craton. There exist few geophysical studies (Kumar et al., 2012; Gokarn et al., 2013; Mishra, 2015) on the Bundelkhand craton. These studies suggested plume/superplume setting was responsible for the formation of the Proterozoic basins of this region and even proposed the existence of an underplated mafic layer below the basins. However, a detailed subsurface model depicting the spatial and depth extent of the underplated layer based on geophysical observation and understanding its correlation with the development of the Proterozoic basins along the southern margin of the Bundelkhand craton and adjoining areas are not available in the literature. As a result, constraints for the gravity modelling are limited and we have utilized the layering information based on the present radially average power spectra analysis as well as incorporated all the available information related to the rock types, and layer thickness as described below.

In the present modelling, the thicknesses for the different layers are majorly constrained by the results from the studies conducted using wide-angle seismic data along the Hirapur-Mandla profile by Sain et al. (2000) and the shear velocity structure given by Kumar et al. (2012), along with the depths as obtained from the radially averaged power spectrum. Best attempts have been made to carefully stick to the prior established density and thickness estimates as mentioned in Table 1. The density and thickness of the underplated layer are modified and adjusted by a trial-and-error approach to fit the gravity response curve, keeping the error between the calculated and observed gravity response as low as possible. The Moho depth for the Sagar station (~44km) suggested by Kumar et al. (2012) has been used as a constraint for the region where the underplated layer is modelled. All these details related to the constraints used for gravity modelling are already described in the section '3.5 Two-dimension forward gravity modelling'.

**Table 1: Density values used in the present study, compiled from established literature.**

| Layers | Density (kg/m$^3$) | References |
|---|---|---|
| Recent sediments | 2100 | Prasad et al. (2018) |
| Vindhyan supergroup | 2500 | Mishra (2015); Pal and Kumar (2019) |
| Bijawar basement of Vindhyan | 2840 | Mishra (2015 |
| Bundelkhand granite + basement, Upper crust (average) | 2640 | Podugu et al. (2017); Pati and Singh (2020) |
| Deccan traps | 2850 | Rao et al. (2011) |
| Average Middle and Lower crustal density | 2800 | Rao et al. (2011); Chouhan et al. (2020) |
| Underplated layer | 3150 | Chouhan et al. (2020) |
| Upper mantle | 3300 | Rao et al. (2011); Chouhan et al. (2020) |

5. *To generalize that the high density anomaly has sources extending from deep to shallow is an over simplification. The authors have done upward continuation to a single height and they base their conclusion on it. Try to upward continue to a height of 60 (corresponding to sources at a depth of 30 km and below) to justify presence of underplate at depths in the order of 30 km in the models. In addition, the central part of the models (central region) do not show the extension of high anomaly sources to shallower depths as shown by the residual anomaly.*

**Response:** We are thankful to the reviewer for pointing out this concern. We apologise for this confusion. We want to clarify here that this conclusion was not based on one upward continued map but was based on a trial-and-error approach with various UC heights as suggested by Gupta and Ramani (1980). However, only the 30 km upward continued regional anomaly and associated residual maps were only included in the manuscript based on visual correlation with regional trend and local features, respectively. We have utilised these results to qualitatively understand the continuation of the high-density sources at different depth level.

We have now included all the results based on the upward continuation heights of 60 km, 30 km, and 10 km (Figs. 3A, 3B, 3C, respectively, see the figures under the response on comment # 4 above; and Fig. 4a-f of the revised manuscript) corresponding to the depth estimates from radially averaged power

spectrum analysis, i.e., ~30.3 km, ~11.9 km, and ~2.7 km, respectively. The 60 km, 30 km, and 10 km upward continued regional anomalies, all showing highs occurring in the SW corner (Figs. 3A(a), 3B(a), and 3C(a), respectively). The regional and residual anomaly maps obtained by the 10 km upward continuation method (Fig. 3C) show similarity to those obtained from the 30 km upward continuation method (Fig. 3B). We have also provided the results obtained from the upward continuation up to 40 km and 50 km heights to further validate the statement referred to in the comment (Figs. 4A and 4B, see below). The centrally located high gravity anomaly, as seen for both 40 km and 50 km upward continued regional anomalies, exhibits that the high gravity signatures due to high-density material are observed at depths shallower than 30 km. These suggest the continuation of high-density sources from deeper to shallower depths. The data used has a resolution of 1-minute and thus the residual signatures generally correspond to depths up to 5-6 km. This implies that the higher gravity anomalies in the central and southwestern regions, as observed in the regional as well as residual anomalies obtained from upward continuation heights of 60 km, 50 km, 40 km, 30 km, and 10 km, are due to sources at deeper and shallower depths (Figs. 3, 4). Based on the geological set up of this region and above observations, it can be inferred that the upwelling magma and eventual magmatic emplacement as underplated layer at shallower depth may have caused such anomaly pattern.

[Figure]

**Figure 4: A. (a) Regional gravity anomaly map of the global gravity data, upward continued up to 40 km, (b) Residual gravity anomaly map of the global grid data, obtained after subtracting the 40 km upward continued regional gravity anomaly from complete Bouguer anomaly. B. (a) Regional gravity anomaly map of the global gravity data, upward continued up to 50 km, (b) Residual gravity anomaly map of the global grid data, obtained after subtracting the 50 km upward continued regional gravity anomaly from complete Bouguer anomaly.**

6. *On Line 332 the paper asserts that there is **striking** similarity between the inverted Moho topography and the gravity signature. Please tone down this assertion. There is similarity but pay attention to the following remarks: a. the gravity anomaly from the inverted Moho topography*

*shows a slightly southward shift and centered at the southern margin of the regional anomaly. b. at the southwestern corner the effect is in fact opposite. There is high anomaly in the both the regional and residual gravity but low anomaly in the gravity obtained from the Moho.*

**Response:** We are thankful to the reviewer for raising this concern as it helped us to remove the confusion. We agree with the reviewer's observation on figures 5b and 3a of the manuscript (original submission). However, these two figures are generated based on two different mathematical concepts but still show significant similarities in terms of anomaly trend and overall pattern. These converging results provide confidence in the interpretation. Maybe we were too optimistic about this, however, as per the suggestion of the reviewer the statement has been toned down.

We also want to mention here that Figures 5b and 3a (of the original manuscript) had different areal coverage as the Parker-Oldenburg inversion scheme (used for Figure 5b of the original manuscript) required square shaped area. As a result, the latitudinal extent of the data used for Figure 5b (of the original manuscript) was 23°–25° N but that for the present study area is 23.5°–25° N. This might have resulted in the observation of point *b* in the comment. To avoid any confusion, the new figure (Fig. 5a, see below; Figure 6a in the revised manuscript) corresponding to the results of the inversion method has the study area marked by a red box, and the colour map format for both figures is now kept as same. The E-W trending high gravity in both inversion results (Fig. 5a) and 30 km upward continued regional anomaly (Fig. 5b, see below) shows a general similarity in the trend.

[Figure]

**Figure 5: (a) Gravity map obtained using the inverted Moho topography obtained from Parker-Oldenburg algorithm. The red box marks the study area as seen in the adjacent regional anomaly map. (b) Regional Bouguer anomaly map of the global gravity data upward continued up to 30 km.**

7. *I have serious reservation on the contradictory results obtained from the two approaches of the gravity modeling. The authors obtained the undulation of the Moho interface using the downward continuation formula as given by the modification of Parker-Oldenburg algorithm (Fig. 5a). As is also noted by the authors the Moho depth undulations obtained by this method beneath the basins where underplate is observed do not correspond to the depths obtained by forward modeling (Fig. 6 & 7). The underplate top is considered as Moho in Fig. 5a whereas the underplate bottom is*

*considered as Moho in both Fig. 6 & 7. In addition, there is also a clear discrepancy in areas where there is no underplate. Compare, for example, Fig. 5a with Fig. 7. In northeast where there is no underplate the Moho depth varies between 37 and 39 km in the forward model in Fig. 7 whereas in Fig. 5a the depth variation for the same area is between 33 and 44 km. These contradictions should not have occurred since there is a clear density contrast between the mantle and the underplate and the Parker-Oldenburg approach is capable of recognizing the difference between them.*

**Response:** We acknowledge the observation made by the reviewer. The average crustal density value of 2.78 g/cm$^3$ is obtained by taking the average densities of the crustal layers corresponding to the Vindhyan sequences, Bijawar group, average density of Upper crust (Bundelkhand granite + basement), average density of middle and lower crust as well as the underplating (all these densities are mentioned in Table 1 of the manuscript and below comment #4). The density contrast then obtained with the mantle (i.e., 0.52 g/cm$^3$) was used for the inversion (Fig. 6A, see below). The Parker-Oldenburg method is also applied using a density contrast of 0.6 g/cm$^3$ (Fig. 6B, see below) using the average crustal density eliminating the underplating density. The results (Figs. 6A,6B) follow similar trends as the results shown in the manuscript, and the Moho values in the central region (below the Vindhyan rocks) show a range of values from 32 km to 36 km. Thus, we observed that a density difference of 0.15 g/cm$^3$ between the underplated layer and mantle is not very distinctly observed in Moho depth variations, irrespective of whether the underplated layer is included in the average crustal density calculations or not. As the inversion results are unable to distinguish the underplated layers, the obtained Moho depths appear shallower in the Parker-Oldenburg inversion method. This can be observed in the Distance vs. Moho depth plot using the inverted Moho topography to understand the Moho trend below the profiles AAꞋ and BBꞋ that were used for the forward models (Figs. 7A and 7B, see below). These plots show that the general trend of the underplating interface from the forward model and the trend of the Moho structure from the inversion results along the profiles AAꞋ (Fig. 7A) and BBꞋ (Fig. 7B) exhibit similarity.

The depths used in the gravity modeling are essentially based on a combination of the results of RAPS analysis, inverted Moho topography, and the crustal layer thicknesses obtained from prior literature (as mentioned in comment #4). The discrepancy pointed out in the Moho depths for the regions not exhibiting the underplating in the forward models (e.g. below the Bundelkhand craton) is possibly observed as they have been constructed by adjusting the layer thicknesses, and depths according to the above-mentioned constraints. The Moho depth ranges for regions covered by the Bundelkhand craton and the Vindhyan basin as suggested by Kumar et al. (2012) have been used as a constraint for the region of the modelled underplated layer, which varies from ~36 km to ~44 km. These depth ranges were used as constraints with a trial-and-error approach for the forward modelling while keeping the density values consistent with the density contrast utilized for the inversion method. Therefore, the forward models represent more refined and better-constrained results which follow the broad trend of inversion results.

[Figure]

**Figure 6: A. (a)** Moho topography map obtained by applying the Parker-Oldenburg method on the Bouguer gravity data of Fig. 2, using 0.52 g/cm³ as the density contrast. **(b)** Gravity map obtained using the inverted Moho depths from Fig. 6A(a). The red box marks the study area. **B. (a)** Moho topography map obtained by applying the Parker-Oldenburg method on the Bouguer gravity data of Fig. 2, using 0.6 g/cm³ as the density contrast. **(b)** Gravity map obtained using the inverted Moho depths from Fig. 6B(a). The red box marks the study area.

[Figure]

**a**

[Figure]

**b**

Figure 7: (a) Distance vs. Moho depth plot using the inverted Moho topography, Moho interface from the forward model, and underplating interface from the forward model along the profile AAʹ. (b) Distance vs. Moho depth plot using the inverted Moho topography, Moho interface from the forward model, and underplating interface from the forward model along the profile BBʹ

TECHNICAL CORRECTIONS

**Response:** The below-mentioned changes have been made in the manuscript.

- On Line 29 "…..attributed to the presence of underplated layers, like rift basins…" Please remove the word "like".

- On Lines 194 and 197- 8 the phrase h(x) and $z_o$ are redundant. Please eliminate the repetition.

- On Line 346 it must be "crustal uplift" and not "crustal upliftment".

- In Fig. 5   contour values can be indicated on every other line to avoid congestion.

**REFERENCES**

Basu, A. and Bickford, M. E. An alternate perspective on the opening and closing of the intracratonic Purana basins in peninsular India, Journal of Geological Society of India 85(1), 5–25, 2015.

Gupta, V. K., and Ramani, N. Some aspects of regional-residual separation of gravity anomalies in a Precambrian terrain. Geophysics 45: 1412-1426, 1980.

Jacobsen, B.H. Case for upward continuation as a standard separation filter for potential-field potential-field maps. Geophysics 52, 1138–1148, 1987.

Kebede, H., Alemu, A., Fisseha, S. Upward continuation and polynomial trend analysis as a gravity data decomposition, case study at Ziway-Shala basin, central Main Ethiopian rift. Heliyon, 6, 2020.

Kumar, T. V., Jagadeesh, S., and Rai, S.S.: Crustal structure beneath the Archean–Proterozoic terrain of north India from receiver function modelling, Journal of Asian Earth Sciences 58, 108–118, 2012.

Meng, X., Guo, L., Chen, Z. Shuling, L., and Lei, Shi. A method for gravity anomaly separation based on preferential continuation and its application. Appl. Geophys. 6, 217–225 2009.

Mishra, D. C.: Plume and Plate Tectonics Model for Formation of some Proterozoic Basins of India along Contemporary Mobile Belts: Mahakoshal — Bijawar, Vindhyan and Cuddapah Basins, Journal of the Geological Society of India 85(5), 525–536, 2015.

Oasis Montaj MAGMAP manual: http://updates.geosoft. com/downloads/Bles/how-to-guides/Getting˙Started˙with˙ montaj˙MAGMAP˙Filtering.pdf.

Pal, S. K. and Kumar, S.: Subsurface Structural Mapping using EIGEN6C4 Data over Bundelkhand Craton and Surroundings: An Appraisal on Kimberlite/lamproite Emplacement. Journal of the Geological Society of India, 94(2), 188–196, 2019.

Sain, K., Bruguier, N., Murty, A. S. N., and Reddy, P. R.: Shallow velocity structure along the Hirapur-Mandla profile using travel time inversion of wide-angle seismic data, and its tectonic implications. Geophysical Journal International 142(2), 505–515, 2000.

**D. Authors' response to the comments from Referee #2**

Note: Figures numbers are given as per the sequence of appearances of the figures in the responses to the comments of reviewer # 2; wherever required corresponding figure number in the revised manuscript is also referred.

1. *The authors use global gravity data and upward continue Bouguer gravity anomaly to investigate the crustal structure within the study area. I understand that upward continuing acts as a low pass filter to enhance the long-wavelength regional Bouguer anomaly trend.*

   *Please, can the authors explain why they chose 30 km elevation for the upward continuation? Is 30 km the elevation at which the effect of surface structures become negligible?*

   *It would be interesting to provide a few test examples of upward continuation at different elevations (possibly as supplementary material); also, see e.g. Zeng et al. (2007), Geophysics, for approaches to estimate ideal elevation for upward continuation.*

   *Also, the terrain correction seems to be small (less than 1 mGal), but topography ranges from 175 m to 617 m (line 152). Can the authors provide a topographic map of the study area, together with the grid points from the global gravity data they used?*

**Response:** We would like to clarify that the choice of the upward continuing heights was based on a trial-and-error approach as suggested by Gupta and Ramani (1980). The 30 km upward continued regional gravity anomaly shows some similarities with the overall trend observed in the Complete Bouguer anomaly map (Fig. 1, below; Fig. 3 of the revised manuscript). Consequently, the corresponding residual gravity anomaly obtained after removing the 30 km upward continued regional gravity anomaly also showed some correlations with the lithological units observed in Figure 1b of the manuscript. Thus, these maps were the ones included in the manuscript and utilized these results to understand the high-density sources causing these signatures at various depths.

We thank the reviewer for their suggestion based on the work of Zeng et al. (2007) where they demonstrated a logical procedure for selection of best upward continued height for regional-residual separation. However, they also concluded that their method also gave ambiguous choice of upward continued heights for separation of gravity anomaly due to multiple sources and there is no optimum height. Based on the theoretical concept, several authors also consider the upward continuation height as twice the value of the source depth (Jacobsen, 1987; Meng et al., 2009; Pal and Kumar, 2019; Kebede et al., 2020). Thus, following the suggestions provided, we have now added the regional-residual gravity anomaly results based on the upward continuation heights of 60 km, 30 km, and 10 km (Figs. 2A, 2B, 2C, respectively, see below; Fig, 4a-f of revised manuscript) approximately corresponding to the depth estimates from radially averaged power spectrum analysis, i.e., ~30.3 km, ~11.9 km, and ~2.7 km, respectively. The 60 km ed upward continued regional gravity anomaly show the moderate to high anomalies trending from the central region up to the SW corner of the study area. The regional and residual gravity anomaly maps obtained by the 10 km upward continuation method (Fig. 2C) show similarity to those obtained from the 30 km upward continuation method (Fig. 2B). These show the signatures from the high-density sources from deeper as well as shallower depths. Therefore, based on the above observations and the geological setup of this region, we infer that the magmatic emplacement as an underplated layer at the lower crustal levels as well as the volcanogenic rock sequences of the Bijawar group at shallower depth (Mishra, 2015) may have caused such anomaly patterns in the upward continued regional gravity anomaly maps and the corresponding residual gravity anomaly maps.

We thank the reviewer for their suggestion here and raising the concern related to the terrain correction value. The topography map for the study area is presented here in the Figure 3a (see Fig. 2 of the

revised manuscript), showing a variation from175 m to 617 m, derived from the global 1-minute topography grids available on the website of the Scripps Institution of Oceanography (Smith and Sandwell, 1997). The terrain correction is calculated and applied using the Terrain feature available on the Geosoft Oasis Montaj software, and on cross-checking with the values derived from the module, the maximum terrain correction obtained is ~ 0.93 mGal. We sincerely apologise for overlooking this. This has now been rectified in the revised manuscript. A plot depicting the terrain correction values calculated over the study area is presented in the Figure 3b. After the application of the terrain correction, the obtained terrain corrected Bouguer anomaly is observed to be similar in trend to the Bouguer anomaly. This can be observed in Figure 4. Since the plotted topographic map is derived from a global 1-minute gridded data, plotting the gravity data points on the topographic map results in a map with densely populated grid points, obscuring the observed topographic variations. Thus, we have avoided plotting the data points on the topographic map.

[Figure]

**Figure 1: Complete Bouguer anomaly map (lithology map from Figure 1b of the revised manuscript superimposed) obtained using topography and gravity data from global 1-minute topography and free-air gravity grids available on the website of the Scripps Institution of Oceanography, (https://topex.ucsd.edu/WWW_html/mar_topo.html; https://topex.ucsd.edu/cgi-bin/get_data.cgi). Locations: (1)Lalitpur, (2)Mungaoli, (3)Khurai, (4)Gyaraspur, (5)Sagar, (6)Banda, (7)Sonrai, (8)Girar, (9)Madawara, (10)Karitoran, (11)Tikamgarh, (12)Chhatarpur, (13)Khajuraho, (14)Bijawar, (15)Dargawan, (16)Hirapur, (17)Hatta, (18)Damoh.**

[Figure]

**Figure 2: A. (a) Regional gravity anomaly map of the global gravity data, upward continued up to 60 km, (b) Residual gravity anomaly map of the global grid data, obtained after subtracting the 60 km upward continued regional gravity anomaly from complete Bouguer anomaly. B. (a) Regional gravity anomaly map of the global gravity data, upward continued up to 30 km, (b) Residual gravity anomaly map of the global grid data, obtained after subtracting the 30 km upward continued regional gravity anomaly from complete Bouguer anomaly. C. (a) Regional gravity anomaly map of the global gravity data, upward continued up to 10 km, (b) Residual gravity anomaly map of the global grid data, obtained after subtracting the 10 km upward continued regional gravity anomaly from complete Bouguer anomaly.**

[Figure]

**Figure 3: (a) Topographic map derived from the global 1-minute topography grids available on the website of the Scripps Institution of Oceanography, (https://topex.ucsd.edu/WWW_html/mar_topo.html; https://topex.ucsd.edu/cgi-bin/get_data.cgi), (b) Plot of terrain correction values calculated using the Terrain feature on the Gravity module on Geosoft Oasis Montaj software.**

[Figure]

a                                                          b

**Figure 4: (a) Bouguer anomaly map without terrain correction applied, (b) Terrain corrected complete Bouguer anomaly map.**

2.  *The authors extract the regional Bouguer anomaly trend and separate it from the gravity effect of surface geological structures (residuals). However, they often refer to "Bouguer gravity anomaly" throughout the manuscript.*
    *Please, state explicitly which type of gravity data product is used for RAPS analysis, 3D Moho depth inversion, and 2D forward gravity modeling respectively.*
    *Is the upward-continued regional Bouguer trend used in every analysis? If yes, can it constrain the shallower (hence smaller wavelength) structures in the 2D forward modeling (e.g. basin structures at a few km depth below the surface in Figures 6 and 7)?*

**Response:** The phrases " Bouguer gravity anomaly" and "complete Bouguer anomaly" have been used to describe the gravity anomaly (Fig. 1, see above under the response to comment #1 and Fig. 3 of revised manuscript) that is derived based on the application of Bouguer and terrain corrections on the free-air anomaly obtained from the gridded global gravity data. This Bouguer anomaly is used for all the analyses described in the manuscript (i.e., for RAPS, 3D Moho depth inversion, and 2D forward gravity modelling). Maybe the confusion arose due to the phrases "regional Bouguer anomaly" and "residual Bouguer anomaly" being used a few times in the manuscript for describing the resultant regional and residual maps (Figs. 3a and 3b, in the original manuscript) from the upward continuation approach.

To avoid further confusion, we have modified the manuscript to denote the terrain corrected Bouguer anomaly as "complete Bouguer anomaly" (Fig. 1, see above under the response to comment #1), while the upward continued regional and calculated residual anomalies as "regional gravity anomaly" and "residual gravity anomaly", respectively.

Once again, we want to clarify here that all the analyses carried out in this study utilize the complete Bouguer anomaly (as given in Fig. 1, see above under the response to comment #1; and in Fig. 3 of the revised manuscript) and this has been mentioned in the Methodology as well as the Results sections. Thus, the shallower structures have been created in the forward models utilizing the complete Bouguer anomaly.

3.  *Could the authors spend a few more words on the RAPS technique? The output provides a deep density contrast interface at ca. 30.3 km depth. What is the resolution expected from this technique, which, if I understand well, provides a 1D average information across the study area?*

**Response:** The section on radially averaged power spectrum (RAPS) analysis in the manuscript is a concise form of explanation for the technique as this is a well-established and commonly used technique for potential field data interpretation. The detailed mathematical background for the technique can be found in one of the pioneer works in this direction by Spector and Grant (1970). In terms of the resolution of the technique, the reviewer's perspective is valid since the technique estimates the average depth to the top of the assemblage of the source bodies. According to Wahaab et al. (2017), deeper sources can be distinguished from the shallower ones only if they have greater spectrum amplitudes or if the shallower bodies have smaller depth extent. The extent of the depth estimates derived from the RAPS technique depends on the spatial dimension as well as resolution of the data being used. According to Spector and Grant (1970), the depth of exploration for this technique depends on the size of the map being used. Thus, owing to the large latitudinal and longitudinal extent (23.5°–25° N and 78°–80° E) of the study area, we were able to obtain the average depth estimate of ~ 30 km, to deeper interfaces. The global gravity data used here has a spatial resolution of 1-minute, which aids in better examination of the deeper features in the present study area.

4.  *The authors perform a 3D inversion for the Moho topography. The algorithm they use requires assuming a mean depth z0 and a density contrast. Why did they use 36 km for z0? And not e.g. 30 km as obtained from the previous RAPS analysis?*
    *And related to this, how do these results change as a function of z0 and density contrast? Please, provide a sensitivity test for these parameters, and a resolution test for the inversion algorithm application.*

**Response:** We appreciate the suggestions provided by the reviewer here. We did perform the 3D inversion for obtaining the Moho topography using varying values of mean Moho depth ($z_0$), ranging from 30 km to 38 km (Fig. 5, as given below). The density contrast (i.e., 0.52 $g/cm^3$) between the mantle and the crust, used for the inversion is obtained by taking the average crustal density value of 2.78 $g/cm^3$ (all these densities are mentioned in Table 1 of the manuscript as well as the same table has been included below). This density contrast was thus kept constant for the varying $z_0$ values. The inverted Moho topography results obtained using 36 km as the mean Moho depth shows closer correlation with the crustal layer thicknesses and depths suggested in the work by Kumar et al. (2012). The average Moho depth of ~36 km for the Archean crust of the Bundelkhand craton was also presented by Kumar et al. (2012), thus we finalized the results with the assumed $z_0$ value of 36 km. The variations observed in the obtained inverted Moho topography by varying the mean depth values is presented here in the Figure 5. The results obtained with $z_0$ values of 30 km, 32 km, and 34 km (Figs 5A, 5B, and 5C, respectively, below), show a very shallow Moho structure below the central region of the study area (ranging from 26 km to 30 km), which is also shallower than the depth estimates to the mafic layer proposed by Kumar et al. (2012). Thus, the results indicate that the estimated Moho depths are changing significantly with 2 km variation in the $z_0$ values. We also noticed that by varying the density contrast between the mantle and the crust from 0.52 $g/cm^3$ (including the density of the underplated layer in crustal part) (Fig. 6A, below, and see Fig. 6a-b in the revised manuscript) to a density contrast of 0.6 $g/cm^3$ (without considering any underplating layer in the crustal part) (Fig. 6B, below), the results qualitatively remain similar. Thus, the algorithm shows little sensitivity to the change in the assumed density contrast.

[Figure]

**Figure 5: Moho topography map obtained by applying the Parker-Oldenburg method on the complete Bouguer anomaly data of Fig. 1, using 0.52 g/cm³ as the density contrast, with A. $z_0$= 30 km, B. $z_0$= 32 km, C. $z_0$= 34 km, D. $z_0$= 36 km, and E. $z_0$= 38 km.**

[Figure]

A          B

**Figure 6: A. (a) Moho topography map obtained by applying the Parker-Oldenburg method on the complete Bouguer gravity data of Fig. 1, using 0.52 g/cm³ as the density contrast. (b) Gravity map obtained using the inverted Moho depths from Fig. 5A(a). The red box marks the study area. B. (a) Moho topography map obtained by applying the Parker-Oldenburg method on the complete Bouguer gravity data of Fig. 1, using 0.6 g/cm³ as the density contrast. (b) Gravity map obtained using the inverted Moho depths from Fig. 5B(a). The red box marks the study area.**

5. *I believe a few important points should be clarified in the 2D modeling stage.*

   *The authors perform 2D forward modeling based on the formulas by Talwani et al. (1959). These formulas are best suited when both the data, and the target area, present a 2.5D symmetry (e.g. changing properties along the x-z plane, and no changes along y-axis); see e.g. Scarponi et al. (2021), Frontiers for one example. In particular, profile AA' seems not to be perpendicular to visible 2.5D structures, nor in the gravity data or in the underlying, inverted Moho structure (based on Figure 2 and Figure 5a). A slight-to-moderate rotation of profile AA' around its center (e.g. +20 degrees) could potentially provide a different gravity data profile, and hence lead to different results and interpretation.*

   a. *The authors could consider using a 3D inversion software (see e.g. IGMAS+ in Spooner et al. 2019, Solid Earth). If not 3D, how would the 2D gravity profiles (data and models) look like along a set of parallel profiles (e.g. at constant longitude)? Would the results, and hence interpretation, change along a different profile than AA'? This should be tested and discussed before interpretation.*

   b. *Paragraph 3.5 on the construction of the 2D profiles never mentions incorporating the results from the 3D Moho depth inversion. Were these Moho results neglected in the creation of the 2D models shown in Figure 6 and 7? If yes, why?*

*The authors mention the RAPS estimate as reference used in the profiles, but RAPS provides an inherently 1D average information. Moreover, profile BB' shows no interfaces around 30 km: please, can the authors explain the reason for this?*

*According to Figure 5a, the computed Moho depths obtained along BB' range from 44 km to 34 km depth, but this seems not to be the case when looking at Figure 7. This should be clarified (partially applies also to profile AA' and Figure 6).*

c. *In the definition of the structures within profiles AA' and BB', the authors refer to a list of previous investigations, to be used as external constraints. This is OK in principle. However, these external constraints are not explicitly indicated in Figure 6 and 7.*

*Which geometries were imported as unmodified external information? Which ones were generated and/or modified by the authors? This information is not clear and should be made explicit. The authors could also show in Figures 6 and 7, how their new Moho estimate compares to the external information they refer to.*

*Clarifying the points above is important to discuss the fit to the gravity data along the selected profiles. For example, was the geometry of the underplating structure in Figure 7 imported from external sources? How would the forward modeling compare to the data, with a different, or without, the underplating layer along AA'?*

*To address these points, I would advise starting by: 1) apply RAPS; 2) use the RAPS deepest interface estimate as z0 for the Moho inversion (provide sensitivity and resolution tests); 3) use the obtained Moho depth as starting geometrical constraint, together with those existing in the literature, either for 3D modeling, or for a set of 2D profiles (as much as possible along structures with 2.5D symmetry), providing support for the chosen 2D profiles; 4) test if the deeper underplating layers can be resolved by gravity along the chosen profiles.*

**Response:** We want to clarify here that we have utilised the GM-SYS profile module of the Oasis Montaj software for performing the 2D forward modelling along the two profiles. As per the GM-SYS User's Guide, the 2D forward modelling responses are based on the methods of Talwani et al. (1959), and Talwani and Heirtzler (1964), that make use of the algorithms given by Won and Bevis (19870). It also performs 2.5D calculations based on the method of Rasmussen and Pedersen (1979). These methods and algorithms are combined to improvise the efficiency of the features of GM-SYS profile module to enable 2¾-D forward modelling (Oasis Montaj GM-SYS manual). Thus, this module helps in developing 2-D forward models, 2¾-D forward models as well as skewed models where the strike direction may not be perpendicular to the profile. The E-W trending (approximately) gravity high in the central region of Figure 1, presented here, is not exactly perpendicular to the two profiles but they intersect the anomaly feature (also the inferred underplated layer) at an angle. Since the GM-SYS profile can be utilized to develop models along such profiles, the chosen profiles were eventually used.

a) We appreciate suggestions provided here by the reviewer. Previously, models were developed along profiles at constant longitudes, but the individual profiles did not span all the lithological units of interest to give a better understanding of the crustal structure below the study area. Hence, the present profiles were utilized for the forward models. The methodology utilized here are chosen to provide validation to the plume/superplume hypothesis suggested by Mishra (2015) by showing the presence of the underplated layer at lower crustal depths. The trend of the Moho structure (as obtained from forward model) and the depth variations observed from the inverted Moho topography show some correlation with the layering information from

Kumar et al. (2012). This can be also observed in the Distance vs. Moho depth plot using the inverted Moho topography (as obtained from Parker-Oldenburg inversion method) along with the Moho depth with and without the thickness of underplated layers as obtained from forward models along the profiles AAʹ and BBʹ (Figs. 7a and 7b, below). These plots show that the general trend of the underplating interface from the forward model and the trend of the Moho structure from the inversion results show similarity along the profiles AAʹ (Fig. 7a) and BBʹ (Fig. 7b). However, the inversion scheme is unable to distinguish the underplated layer from the Moho depth level unlike the forward modeling scheme. Therefore, the forward models represent more refined and better-constrained results which follow the broad trend of inversion results.

We are grateful to the reviewer for their suggestion of the IGMAS+ software and we will attempt to accommodate results from it in future work.

[Figure]

**A**

[Figure]

**B**

**Figure 7: (a) Distance vs. Moho depth plot using the inverted Moho topography, Moho interface from the forward model, and underplating interface from the forward model along the profile AAʹ. (b) Distance vs. Moho depth plot using the inverted Moho topography, Moho interface from the forward model, and underplating interface from the forward model along the profile BBʹ**

b) We acknowledge the observation made by the reviewer here. However, we want to clarify that the depths used in the gravity modeling are a combination of the results obtained from RAPS

analysis, inverted Moho topography, and the crustal layer thicknesses obtained from prior literature. The average depth estimate of 30 km to the deepest interface as per the RAPS analysis helped in constraining the depth to the top of the underplated layer. The depth to top of the underplated layer corresponds to the region showing shallowest Moho depths in the inverted Moho topography results, which corroborates with the findings of Kumar et al. (2012) and hence is used as a constraint too. The difference in the Moho depths for the regions in the forward models and the inverted Moho topography below the Bundelkhand craton is possibly observed as the forward models have been constructed by adjusting the layer thicknesses, and depths according to the above-mentioned constraints. Also, the density contrast (0.52 g/cm$^3$, with the density of the underplated layer, Fig. 6A, above) used for the inversion may not apply objectively for the region below Bundelkhand craton since it is most absent below the craton. As briefly discussed in the response to comment #4 above, we performed the Moho depth inversion with the density contrast (between Mantle and crust) values of 0.52 g/cm$^3$ (using the average crustal density including the underplating density, Fig. 6A, above) and 0.6 g/cm$^3$ (using the average crustal density eliminating the underplating density, Fig. 6B, above). The results (Figs. 6A,6B) follow similar trends as the results shown in the manuscript, and the Moho values below the Bundelkhand craton and in the central region (below the Vindhyan rocks) show similar ranges of values. Thus, we observed that the inversion results are unable to distinguish the density contrast due to the underplated layer. The Moho depth ranges for regions covered by the Bundelkhand craton and the Vindhyan basin as suggested by Kumar et al. (2012) have been used as a constraint for the region of the modelled underplated layer, which varies from ~36 km to ~44 km. These depth ranges were used as constraints with a trial-and-error approach for the forward modelling while keeping the density value consistent with the density contrast utilized for the inversion method. Thus, the forward models represent more refined and better-constrained results which follow the broad trend of inversion results.

c) Due to the non-availability of sufficient studies over the present study area, along the southern margin of Bundelkhand craton, the gravity modelling is constrained with only limited information. Few of the existing geophysical studies (Kumar et al., 2012; Gokarn et al., 2013; Mishra, 2015) on the Bundelkhand craton have suggested plume/superplume setting one of the formation mechanisms of the Proterozoic basins of this region and even proposed the presence of an underplated mafic layer below the basins. There is a lack of a detailed subsurface model delineating the spatial and depth extent of the underplated layer based on geophysical observation and its correlation with the formation of the Proterozoic basins along the southern margin of the Bundelkhand craton and adjoining areas. For the forward modelling here, the thicknesses for the different layers are majorly constrained by the results from the wide-angle seismic study along the Hirapur-Mandla profile by Sain et al. (2000) and the shear velocity structure obtained by Kumar et al. (2012). Best attempts have been made to carefully stick to the prior established density values as mentioned in Table 1. The density and thickness of the underplated layer are modified and adjusted by a trial-and-error approach to fit the gravity response curve, keeping the error between the calculated and observed gravity response as low as possible. The Moho depth for the Sagar station (~44km) suggested by Kumar et al. (2012) has been used as a constraint for the region where the underplated layer is modelled but cannot be displayed on the forward model since it is not directly lying on the BB′ profile. The geometry of layers forming the basin structure is modified using the information and models provided by Basu and Bickford (2015), Mishra (2015). The structure of the underplated is essentially adjusted by the previously addressed trial-and-error approach, with inputs from Kumar et al. (2012) as well as the Moho inversion results. As discussed in the response to comment #5a, it is observed in Figure 7 (above) that the general trend of the top surface of the underplated layer from the forward model and the trend of the Moho structure from the

inversion results show similarity beneath the profiles AAʹ (Fig. 7a) and BBʹ (Fig. 7b). All these details related to the past tectonic evolution information and the constraints used for gravity modelling are discussed in the sections '2 Geological background' and '3.5 Two-dimension forward gravity modelling', respectively.

**d)   Table 1: Density values used in the present study, compiled from established literature.**

| Layers | Density (kg/m$^3$) | References |
|---|---|---|
| Recent sediments | 2100 | Prasad et al. (2018) |
| Vindhyan supergroup | 2500 | Mishra (2015); Pal and Kumar (2019) |
| Bijawar basement of Vindhyan | 2840 | Mishra (2015 |
| Bundelkhand granite + basement, Upper crust (average) | 2640 | Podugu et al. (2017); Pati and Singh (2020) |
| Deccan traps | 2850 | Rao et al. (2011) |
| Average Middle and Lower crustal density | 2800 | Rao et al. (2011); Chouhan et al. (2020) |
| Underplated layer | 3150 | Chouhan et al. (2020) |
| Upper mantle | 3300 | Rao et al. (2011); Chouhan et al. (2020) |

We highly appreciate the suggestions provided by the reviewer on the tentative steps to be carried out. We would like to clarify that the workflow of our chosen methodology follows a similar approach as suggested by the reviewer here, and consequently have presented the results from RAPS analysis, inversion, and forward modelling. We have incorporated all the suggestions put forth by the reviewer to justify our findings as well as refine the existing manuscript for publication.

**Response to additional comments:**

- *Figure 1b is not readable and should be larger. Figure 1a is readable, but please consider using different colors to highlight the different geological units. The perimeter box of figure 1a should appear in figure 1b to show its location;*

**Response:** We apologize if the Figure 1 caused a confusion between Figure 1a and Figure 1b. We have modified the Figure 1 of the manuscript (here given below as Fig. 8, and Fig. 1a-b in the revised manuscript) and the location of the study area is now shown in Figure 1a as a red box. The pattern scheme has been utilized for highlighting the geological units so that this figure can be superimposed with ease on the anomaly maps.

[Figure]

**Figure 8: (a) Position of the Bundelkhand craton and Vindhyan basin with respect to other major cratons of the Indian subcontinent. Bijawar basin forms the base of the Vindhyan basin and the exposed sequences are shown in Figure 1b. (b) General geological setup of the region used for the regional scale study of the craton and surrounding areas along the southern boundary of the craton. The two profiles used for gravity modelling are marked here as AA′ and BB′.**

- *Figure 5 should at least contain a residual map (synthetics minus observations).*
  *It would be also beneficial to add a plot for RMS versus iteration number, to show the RMS reduction during the inversion, and a sensitivity and resolution tests (possibly in a different Figure)*

**Response:** We hope here, by residual the reviewer is indicating the difference between the observed gravity anomaly and the calculated gravity from the inverted Moho topography and the same has be given in Figure 9a (below). The convergence criterion of 0.02 is reached by the $3^{rd}$ iteration with the RMS error value of 0.012106, while performing the inversion with $z_0$ value of 36 km and density contrast of 0.52 $g/cm^3$.

The Parker-Oldenburg algorithm-based Moho depth inversion scheme was introduced by Gomez-Oritz and Agarwal (2005), and it is a well-established approach. Detailed resolution and sensitivity tests are not performed here, like some of the works using this approach done by previous authors (e.g., van der Meijde et al., 2013; Windhari and Handayani, 2015; Bessoni et al., 2020; Ydri et al., 2020). However, as per the reviewer's comments we examined the sensitivity of the inversion scheme with respect to the varying mean depth $z_0$ and density contrast as discussed in the response for the comment #4 above.

[Figure]

**Figure 9: Difference in Gravity between the complete Bouguer anomaly map (Fig. 1, above under the response for comment#1, Rviewer#2) and that obtained using the inverted Moho topography derived from Parker-Oldenburg algorithm, for $z_0$=36 km and density contrast of 0.52 g/cm$^3$. The red box marks the study area.**

- *Figure 6 and 7 should show explicitly which geometries were imported as unmodified external constraints for the construction of the models. They should also show the Moho depth as obtained from the 3D inversion (Figure 5a).*

**Response:** We have clarified the same concern raised by the reviewer in the comment #5c above. To show the trends followed by the Moho structure in the forward models as well as the inversion results, we have provided the plots of Distance vs Depth to Moho along the profiles AAʹ and BBʹ (Figs. 7a and 7b, respectively above).

- *Also, the top banner in Figures 6 and 7 is not very clear: is it gravity or Bouguer anomaly? Please, plot the error on a separate independent scale to be more readable.*

**Response:** These modifications have been incorporated in the modified manuscript, see figures 7 & 8 of the revised manuscript.

- *line 175: by "Bouguer anomaly" you mean the upward-continued regional trend? Please, specify. Same for line 219, 239, 270, 286, 295 and so on.*

**Response:** We have attempted to clarify the similar concern raised by the reviewer in the comment #2 above. Bouguer anomaly always refer to the gravity anomaly caused by the subsurface anomalous masses/structures and include the response of both shallower and deeper features. On the other hand, the upward-continued regional trend represent the response due to deeper features i.e., regional component of Bouguer anomaly. Therefore, Bouguer anomaly cannot never be like the upward-continued regional map. To avoid any confusion (if there), we have modified the manuscript to denote the terrain corrected Bouguer anomaly as "complete Bouguer anomaly" (Fig. 1, see above under the response to comment #1), while the upward continued regional and calculated residual anomalies as "regional gravity anomaly" and "residual gravity anomaly", respectively.

- *Line 244: Does GMSYS perform 2D forward modeling or 2D inversion?*
  *This is a crucial detail. If it performs inversion, then more information is needed here. Or, have you tested several candidate profiles? Please, explain.*

**Response:** The GM-SYS profile module is only used for 2D forward modelling in this study here, however this module is capable of inversion too, which has not been used here.

- *Line 375-377: To my understanding, you obtain a Moho from 3D gravity inversion. But you do NOT obtain a Moho depth from 2D forward gravity modeling (see also question above). If you do not perform 2D inversion, then 2D forward modeling can only validate a certain profile, but not "provide" or "obtain" from it. This is better formulated later in the conclusions, at line 442 "[...] validated by the 2D [...]". Please, This should be clarified. And finally, why not using the 3D Moho results in the construction of the 2D models?*

**Response:** We are thankful to the reviewer for raising this concern, and the concerned statements has been modified, since the forward models validate the presence of the underplated layer, and the Moho structure as suggested by Kumar et al. (2012). We have already clarified part of this query in the response to the comment #5b (above) regarding the use of Moho inversion results in the development of the forward models.

**REFERENCES**

Basu, A. and Bickford, M. E. An alternate perspective on the opening and closing of the intracratonic Purana basins in peninsular India, Journal of Geological Society of India 85(1), 5–25, 2015.

Bessoni, T. P., Bassrei, A., and de Oliveira, L. G. S.: Inversion of satellite gravimetric data from Recôncavo-Tucano-Jatobá Basin System, Brazilian Journal of Geology 50(3), 1–14, 2020.

Gao, X. and Sun, S.: Comment on "3DINVER.M: A MATLAB program to invert the gravity anomaly over a 3D horizontal density interface by Parker-Oldenburg's algorithm", Computers & Geosciences 127, 133–137, 2019.

Gómez-Ortiz, D. and Agarwal, B. N. P.: 3DINVER.M: A MATLAB program to invert the gravity anomaly over a 3D horizontal density interface by Parker-Oldenburg's algorithm. Computers and Geosciences, 31(4), 513–520, 2005.

Gupta, V. K., and Ramani, N. Some aspects of regional-residual separation of gravity anomalies in a Precambrian terrain. Geophysics 45: 1412-1426, 1980.

Jacobsen, B.H. Case for upward continuation as a standard separation filter for potential-field potential-field maps. Geophysics 52, 1138–1148, 1987.

Kebede, H., Alemu, A., Fisseha, S. Upward continuation and polynomial trend analysis as a gravity data decomposition, case study at Ziway-Shala basin, central Main Ethiopian rift. Heliyon, 6, 2020.

Kumar, T. V., Jagadeesh, S., and Rai, S.S.: Crustal structure beneath the Archean–Proterozoic terrain of north India from receiver function modelling, Journal of Asian Earth Sciences 58, 108–118, 2012.

Meng, X., Guo, L., Chen, Z. Shuling, L., and Lei, Shi. A method for gravity anomaly separation based on preferential continuation and its application. Appl. Geophys. 6, 217–225 2009.

Mishra, D. C.: Plume and Plate Tectonics Model for Formation of some Proterozoic Basins of India along Contemporary Mobile Belts: Mahakoshal — Bijawar, Vindhyan and Cuddapah Basins, Journal of the Geological Society of India 85(5), 525–536, 2015.

Oasis Montaj GMSYS manual: http://updates.geosoft.com/ downloads/Bles/how-to-guides/GM-SYS˙ProBle˙Create˙ New˙Model.pdf.

Pal, S. K. and Kumar, S.: Subsurface Structural Mapping using EIGEN6C4 Data over Bundelkhand Craton and Surroundings: An Appraisal on Kimberlite/lamproite Emplacement. Journal of the Geological Society of India, 94(2), 188–196, 2019.

Rasmussen, R., and Pedersen, L. B.: End corrections in potential field modeling: Geophysical Prospecting, 27, 749-760, 1979.

Sain, K., Bruguier, N., Murty, A. S. N., and Reddy, P. R.: Shallow velocity structure along the Hirapur-Mandla profile using travel time inversion of wide-angle seismic data, and its tectonic implications. Geophysical Journal International 142(2), 505–515, 2000.

Smith, W. H. F., and Sandwell, v D. T.: Global seafloor topography from satellite altimetry and ship depth soundings, Science, v. 277, p. 1957-1962, 26, 1997.

Spector, A. and Grant, F. S.: Statistical methods for interpreting aeromagnetic data, Geophysics 35, 293–302, doi:10.1190/1.1440092, 1970.

Talwani, M., Worzel, J. L., and Landisman, M.: Rapid gravity computations for two- dimensional bodies with application to the Mendocino submarine fracture zone: J. Geophys. Res., 64, 49-59, 1959.

Talwani, M., and Heirtzler, J. R.: Computation of magnetic anomalies caused by two- dimensional bodies of arbitrary shape, in Parks, G. A., Ed., Computers in the mineral industries, Part 1: Stanford Univ. Publ., Geological Sciences, 9, 464-480, 1964.

Van der Meijde, M., Julià, J., and Assumpção, M.: Gravity derived Moho for South America. Tectonophysics 609, 456–467, 2013.

Wahaab, F. A, Lawal, S. K., Adebayo, L. L.: Spectral Analysis of Higher Resolution Aeromagnetic Data over Some Part of Kwara State, Nigeria., International Journal of Engineering Research & Technology (IJERT) Volume 06, Issue 03, 2017.

Windhari, A. and Handayani, G.: Gravity data inversion to determine 3D topographical density contrast of Banten area, Indonesia based on fast Fourier transform, In AIP Conference Proceedings (Vol. 1656), American Institute of Physics Inc., 2015.

Won, I. J., and Bevis, M.: Computing the gravitational and magnetic anomalies due to a polygon: Algorithms and Fortran subroutines: Geophysics, 52, 232-238, 1987.

Ydri, A., Idres, M., Ouyed, M., and Samai, S.: Moho geometry beneath northern Algeria from gravity data inversion, Journal of African Earth Sciences, 168, 2020.

**E. Authors' changes in the manuscript**
- The Figure 1 in revised manuscript is modified as per the suggestions by referee #1.
- Lines 154-155 are modified as the topographic map is included in the revised manuscript (Fig. 2, in revised manuscript).
- We have modified the manuscript to denote the terrain corrected Bouguer anomaly as "complete Bouguer anomaly" (Fig. 3, in revised manuscript), while the upward continued regional and calculated residual anomalies as "regional gravity anomaly" and "residual gravity anomaly", respectively.
- Lines 29-30 are modified as per the suggestion of referee #1.
- Lines 126-129 are modified to clarify the location of Bijawar basin.
- The correct terrain correction value is modified in the revised manuscript (Line 157).
- Section 3.2 is modified accordingly as the 60 km, 30 km, and 10 km upward continued regional-residual anomaly maps (Fig. 4, in revised manuscript) are included in the revised manuscript.
- Lines 250-253 are modified to clarify the modelling based on the GM-SYS 2D Profile module.
- Lines 260-262 are modified to specify the constraints used for the 2D forward models.
- Section 4.1 is modified accordingly to discuss the results of the 60 km, 30 km, and 10 km upward continued regional-residual anomaly maps (Fig. 4, in revised manuscript).
- Section 4.2 is modified to incorporate the changes for the results of the 60 km, 30 km, and 10 km upward continued regional-residual anomaly maps with respect to the RAPS depth estimates.
- The Figure 6 is modified as per the suggestions of referee #1, and the section 4.3 is modified accordingly.
- Lines 350-358 are modified accordingly for the results of the 60 km, 30 km, and 10 km upward continued regional-residual anomaly maps (Fig. 4, in revised manuscript).
- Lines 408-419 are modified to provide clarification for comment #7 of referee #1 and comment #4 of referee #2.
- Table 1 is modified to include the density information of the underplated layer.
- The figure numbers and captions are modified as per the new modified figures included in the revised manuscript.

---

## Referee Report (RR1)

**Manuscript: egusphere-2023-1389**
**2nd Report**

I thank the authors for addressing the comments provided in the first round of the revision process, improving the manuscript quality and clarity as well.

I would like to iterate with minor comments on some points. I favor the publication of the manuscript, provided that these points are addressed:

- Line 157: Classical terrain correction calculations usually take into account all the topographic masses within a 166.7 km radius around each gravity observation point (e.g. Zahorec et al. 2021). Is it the same distance used for your terrain calculations? Please, specify if possible.

- Line 231: Please, report the outcome of your sensitivity tests with respect to z0 and density contrast for the Parker-Oldenburg algorithm, and reformulate the sentence. The tests show that the outcome strongly depends on the choice of z0, while the Moho shape is better constrained. Hence, your z0 assumption does not "also correspond with" the Moho depth estimate after Kumar et al. (2012), but you need to assume the Moho depth after Kumar et al. (2012) as z0, to constrain the results of the inversion.

- Line 323: remove "approximately".

- Line 484 and/or Discussion: If I understand correctly, the 30 km depth estimate to the top of a deep dense structure from RAPS, is consistent with the outcome of the Moho 3D inversion above (provided the assumption of z0 after Kumar et al. 2012). However, these two methods are not able to resolve the discontinuity between the mantle below and the crustal underplated layer above, given the weaker density contrast.
  This means that, apart from literature review, the existence of the underplated layer relies only on the outcome of the 2D forward modeling.

  Please, either in the discussion or in the conclusion, add a clear sentence on the limitations of this 2D forward modeling approach: the 2D forward modeling shows that the observed gravity data is compatible with the existence of an underplated layer above the Moho, although it cannot quantify the uniqueness of the proposed solution. An inversion framework will be necessary to assess the uniqueness of the solution proposing such a structure.

---

## Author Response (AR2)

**RESPONSE TO THE EDITOR AND THE REFEREES**

We, the authors, would like to extend our sincere gratitude to the editor and the referees for their valuable time and efforts invested in reading the manuscript. We appreciate and acknowledge all the comments, feedback, and constructive suggestions provided for the further revision of the manuscript. We have responded to each of the comments with the best possible clarifications, as well as considering the feedback, we have carefully incorporated the required changes based on the suggestions of the reviewers in the revised manuscript. The responses to each of the specific comments and the corresponding figures for the clarifications are compiled below.

**A. Responses to the comments from the Topic editor:**

*The authors are requested to take into account every reviewer's comment and address them properly. Particular attention needs to be paid to the sensitivity test and the way in which Zo was chosen. As it is a decisive parameter in determining the Moho depth, it cannot be taken using assumption (Line 188). Did the results of Spector and Grant's (1970) method reflected the mean depth to the Moho? If not why?*

*Additional private note (visible to authors and reviewers only): The authors are requested to accommodate the comments proposed by the reviewers and resubmit the manuscript.*

**Author's Response:** We are thankful to the topic editor for summarizing the major comments of the reviewers thereby pinpointing the issues that need to be tackled in this revision. Accordingly, we have modified the text of the concerned sections in the present revised version of the manuscript.

We want to highlight here that the choice of the mean Moho depth ($z_0$) value for performing the Moho depth inversion was based on a trial-and-error approach with a series of $z_0$ values (at 2km interval) within the range as obtained from the present Radially averaged power spectral analysis and literature derived Moho depth of the region. As a result, we have now reframed our statement (Lines 230-233, in the present revised manuscript) by specifying the range of the mean Moho depth values utilized. Further to enhance the clarity, we have reported the sensitivity of the algorithm with respect to the variation in the $z_0$ values (Lines 236-239) and thereafter, indicated the rationale for selecting the mean Moho depth value of 36 km.

Spector and Grant (1970) noted that the average depth of the buried sources and the size of the map have influence over the power spectrum obtained for an ensemble of sources. It paved the way to show that the logarithm of the radially averaged power spectra of gridded magnetic anomaly data has several constant-slope segments related to statistical ensembles of sources, or equivalent source layers, at different depths. Since then, researchers commonly used this power spectrum-based approach to obtain a first-hand information on the average depth to the top of the source bodies from potential field data (e.g., Nabighian et al., 2005; Chouhan et al. 2020; Sahoo and Pal; 2021 etc.). These depth information act as the initial model for sophisticated inverse modelling and play an important role for applying constraints on different modelling approaches. In the present study, the deepest depth estimate of 30.3 km is used as the starting value of the mean Moho depth parameter in the inversion algorithm. The respective inverted Moho topography results deciphered shallower Moho than expected for this study area as observed in the previous literature. Thus, the depth estimates obtained for this study from

the method given by Spector and Grant (1970) serve as a preliminary constraint for the further approaches like Moho depth inversion and forward modelling.

We have attempted to address and accommodate each of the comments as well suggestions presented by the reviewers to the best of our understanding. The necessary modifications are also incorporated in the text of the revised manuscript.

**B. Responses to the comments from Reviewer 2:**

*I thank the authors for addressing the comments provided in the first round of the revision process, improving the manuscript quality and clarity as well.*
*I would like to iterate with minor comments on some points. I favor the publication of the manuscript, provided that these points are addressed:*

**Author's Response:** We are thankful to Reviewer #2 for appreciating out work and continuously encouraging us towards improving the overall quality of the manuscript through constructive criticisms and valuable suggestions. We have logically replied to all the points raised here and accordingly incorporated the required changes in the text of the revised manuscript.

● *Line 157: Classical terrain correction calculations usually take into account all the topographic masses within a 166.7 km radius around each gravity observation point (e.g. Zahorec et al. 2021). Is it the same distance used for your terrain calculations? Please, specify if possible.*

**Author's Response:** We appreciate the observations made by the reviewer. The terrain correction is applied on the Bouguer corrected gravity anomaly using the 'Terrain correction' module available within the Gravity menu on the Oasis Montaj Geosoft software. The module by default takes the 'Local correction distance' as half of the size of the local DEM grid provided while computing the terrain correction on the software. This distance would approximately be 166.5km (as the local DEM was of 3°×3° dimensions, centred about the present study area), as per the requirements by this module (geosoft gx gravity terrain correction). So, the distance used for terrain calculations was considered by the software as default based on the local DEM grid dimensions. We have indicated our chosen module for the calculation of the Terrain correction in the present revised manuscript (Lines 157-158).

● *Line 231: Please, report the outcome of your sensitivity tests with respect to z0 and density contrast for the Parker-Oldenburg algorithm, and reformulate the sentence. The tests show that the outcome strongly depends on the choice of z0, while the Moho shape is better constrained. Hence, your z0 assumption does not "also correspond with" the Moho depth estimate after Kumar et al. (2012), but you need to assume the Moho depth after Kumar et al. (2012) as z0, to constrain the results of the inversion.*

**Author's Response:** Following the suggestion of the reviewer, we have restructured the statement in Lines 230-233 to report the basis for the range of mean Moho depth values for performing the inversion algorithm. We have also specified the range of the mean Moho depth values utilized here based on the RAPS depth estimates in this study and prior literature. The sensitivity of the algorithm to variation in the $z_0$ values is now reported in the modified

manuscript (Lines 236-239). Based on the suggestion by the reviewer we have now stated the rationale for selecting the mean Moho depth value of 36 km, after the correlations of the respective inversion results and the observations of Kumar et al. (2012).

● *Line 323: remove "approximately".*

**Author's Response:** This has been addressed in the Line 328, in the modified manuscript.

● *Line 484 and/or Discussion: If I understand correctly, the 30 km depth estimate to the top of a deep dense structure from RAPS, is consistent with the outcome of the Moho 3D inversion above (provided the assumption of z0 after Kumar et al. 2012). However, these two methods are not able to resolve the discontinuity between the mantle below and the crustal underplated layer above, given the weaker density contrast.*
*This means that, apart from literature review, the existence of the underplated layer relies only on the outcome of the 2D forward modeling.*
*Please, either in the discussion or in the conclusion, add a clear sentence on the limitations of this 2D forward modeling approach: the 2D forward modeling shows that the observed gravity data is compatible with the existence of an underplated layer above the Moho, although it cannot quantify the uniqueness of the proposed solution. An inversion framework will be necessary to assess the uniqueness of the solution proposing such a structure.*

**Author's Response:** We agree with the observation made by the reviewer with respect to the existence of under plating layer and outcome of forward modelling. Following the suggestion by the reviewer, we have now included a statement observing this limitation (Lines 426-430) in the Discussion part of the modified manuscript.

**C. Responses to the comments from Reviewer 1:**

*Minor technical corrections:*

*1. Figure 1a is of poor quality. the characters are not readable.*

*2. Gupta and Ramani (1980) is indicated in the text but is missing in the reference list.*

**Author's Response:** We are thankful to Reviewer #1 for expressing utmost interest in our manuscript and always providing constructive suggestions. We have incorporated the above two technical corrections in the revised manuscript as described below:

1. The Figure 1a has been modified by increasing the font size as well as incorporating abbreviations for the geological features. The abbreviations are explained in the caption of the figure (Lines 714-716).
2. The missing reference for the mentioned citation has now been included in the modified manuscript (Line 580).

**References**

Chouhan, A. K., Choudhury, P., and Pal, S. K.: New evidence for a thin crust and magmatic underplating beneath the Cambay rift basin, Western India through modelling of EIGEN-6C4 gravity data, Journal of Earth System Science 129(1), doi:10.1007/s12040-019-1335-y, 2020.

Nabighian, M. N., Grauch, V. J. S., Hansen, R.O., LaFehr, T. R., Li, Y, Peirce, J. W., Phillips, J. D., Ruder, M. E.: The historical development of the magnetic method in exploration. Geophysics 70(6):33–61, 2005

Sahoo, S.D., Pal, S.K. Crustal structure and Moho topography of the southern part (18° S–25° S) of Central Indian Ridge using high-resolution EIGEN6C4 global gravity model data. Geo-Mar Lett 41, 3, https://doi.org/10.1007/s00367-020-00679-z, 2021.

Spector, A. and Grant, F. S.: Statistical methods for interpreting aeromagnetic data, Geophysics 35, 293–302, doi:10.1190/1.1440092, 1970.

Terrain correction module information for Oasis Montaj Geosoft: https://help.seequent.com/Oasismontaj/2023.1/Content/gxhelp/g/geosoft_gx_gravity_terrain_correction.htm